# ROCK signaling promotes collagen remodeling to facilitate invasive pancreatic ductal adenocarcinoma tumor cell growth

Nicola Rath[1], Jennifer P Morton[1], Linda Julian[1], Lena Helbig[1], Shereen Kadir[1], Ewan J McGhee[1], Kurt I Anderson[1], Gabriela Kalna[1], Margaret Mullin[2], Andreia V Pinho[3], Ilse Rooman[4], Michael S Samuel[5] & Michael F Olson[1,6,*] (ID)

## Abstract

Pancreatic ductal adenocarcinoma (PDAC) is a major cause of cancer death; identifying PDAC enablers may reveal potential therapeutic targets. Expression of the actomyosin regulatory ROCK1 and ROCK2 kinases increased with tumor progression in human and mouse pancreatic tumors, while elevated ROCK1/ROCK2 expression in human patients, or conditional ROCK2 activation in a $Kras^{G12D}/p53^{R172H}$ mouse PDAC model, was associated with reduced survival. Conditional ROCK1 or ROCK2 activation promoted invasive growth of mouse PDAC cells into three-dimensional collagen matrices by increasing matrix remodeling activities. RNA sequencing revealed a coordinated program of ROCK-induced genes that facilitate extracellular matrix remodeling, with greatest fold-changes for matrix metalloproteinases (MMPs) $Mmp10$ and $Mmp13$. MMP inhibition not only decreased collagen degradation and invasion, but also reduced proliferation in three-dimensional contexts. Treatment of $Kras^{G12D}/p53^{R172H}$ PDAC mice with a ROCK inhibitor prolonged survival, which was associated with increased tumor-associated collagen. These findings reveal an ancillary role for increased ROCK signaling in pancreatic cancer progression to promote extracellular matrix remodeling that facilitates proliferation and invasive tumor growth.

**Keywords** collagen remodeling; extracellular matrix; pancreatic cancer; ROCK kinases; tumor cell invasion

**Subject Categories** Cancer; Digestive System

## Introduction

Patient survival from pancreatic cancer is the lowest of all common cancers, with 5-year survival rates below 5% in England for both men and women. Despite significant effort focused on developing pancreatic cancer targeted therapies, survival rates have not improved, emphasizing the need for additional therapeutic targets and treatment strategies.

The predominant cancer form is pancreatic ductal adenocarcinoma (PDAC), characterized by dense desmoplasia with extensive myofibroblast proliferation and extracellular matrix (ECM) deposition, largely composed of bundled collagen fibers (Chu et al, 2007). The contribution of tumor-associated desmoplasia to PDAC growth and progression is unclear and controversial (Rath & Olson, 2016). The dense stroma serves as a barrier that impairs drug uptake (Feig et al, 2012); acute depletion of tumor-associated stroma using a Hedgehog pathway inhibitor (Olive et al, 2009); or enzymatic degradation of hyaluronic acid (Provenzano et al, 2012; Jacobetz et al, 2013) in mouse PDAC models increased drug uptake and promoted survival. Long-term inhibition or genetic deletion of Hedgehog signaling reduced stromal content and promoted tumor aggression in preclinical mouse models, while Hedgehog pathway activation increased desmoplasia and reduced epithelial cell proliferation (Lee et al, 2014; Rhim et al, 2014). Furthermore, depletion of α-smooth muscle actin (αSMA) expressing myofibroblasts resulted in invasive, undifferentiated tumors with reduced PDAC mouse survival, an observation paralleled by the association between fewer myofibroblasts and worse human patient survival (Ozdemir et al, 2014). Similarly, high PDAC stromal density has been linked with better patient survival (Bever et al, 2015). These observations are consistent with PDAC desmoplasia restraining tumor growth. However, it was reported recently that, although total and fibrillary collagen did not differ significantly in tumors at varying differentiation stages,

1  Cancer Research UK Beatson Institute, Glasgow, UK
2  Electron Microscopy Facility, School of Life Sciences, University of Glasgow, Glasgow, UK
3  Cancer Research Program, The Kinghorn Cancer Centre, Garvan Institute of Medical Research, Sydney, NSW, Australia
4  Oncology Research Centre, Free University Brussels (VUB), Brussels, Belgium
5  Centre for Cancer Biology, SA Pathology and the University of South Australia, Adelaide, SA, Australia
6  Institute of Cancer Sciences, University of Glasgow, Glasgow, UK
   *Corresponding author. Tel: +44 141 330 3654; Fax: +44 141 942 6521; E-mail: m.olson@beatson.gla.ac.uk

collagen fiber diameters increased adjacent to poorly differentiated PDAC tumors, which was associated with short patient survival (Laklai *et al*, 2016). In addition, elevated levels of the collagen cross-linking enzyme lysyl oxidase (LOX) or fibrillar collagen were associated with reduced PDAC patient survival (Miller *et al*, 2015). These seemingly contradictory results illustrate the complex role of PDAC desmoplasia with the possibility of both negative and positive effects on tumor growth and progression, suggesting that the extent of desmosplasia alone is not directly responsible for tumor aggressiveness. By extension, properties that enabled PDAC cells to overcome potential inhibitory constraints on tumor growth imposed by the desmoplastic microenvironment would likely be positively selected, particularly in advanced tumors with well-established stromal components.

Genomic analysis of pancreatic cancers identified core drivers including *KRAS*, *TP53*, *SMAD4,* and *CDKN2A* mutations as well as copy number variations including amplifications of *MET* and *NOTCH1* (Waddell *et al*, 2015; Bailey *et al*, 2016). The *ROCK1* locus on chromosome 18 was found to be amplified in 15% of pancreatic patient tumors (Biankin *et al*, 2012), which was accompanied by concordant copy number/gene expression changes (Bailey *et al*, 2016). The Rho GTPase-regulated ROCK1 and ROCK2 kinases control actomyosin contractility through phosphorylation of substrates including LIM kinases 1&2 (LIMK), myosin-binding subunit of the MLC phosphatase (MYPT1), and regulatory myosin light chain 2 (MLC2) (Rath & Olson, 2012; Julian & Olson, 2014). What role ROCK-mediated actomyosin contractility might play in PDAC has not been established, nor has it been determined whether ROCK2 expression is altered in pancreatic cancer.

In this study, ROCK2 expression was found to increase with pancreatic cancer progression in human and *Kras*[G12D]-driven mouse tumors. Elevated *ROCK1* and/or *ROCK2* expression was associated with shorter survival in human pancreatic cancer patients, while conditional ROCK activation in *Kras*[G12D]-driven PDAC mice was sufficient to accelerate mortality. Conditional ROCK1 or ROCK2 activation in *Kras*[G12D]/*p53*[+/−] mouse PDAC tumor cells promoted collective invasion and proliferation in three-dimensional collagen matrices. Transcriptome analysis identified ROCK-induced differentially expressed gene networks associated with cell adhesion and cell–matrix interactions, with most highly induced transcripts being *Mmp10* and *Mmp13*. ROCK-induced collagen degradation, collective invasion, and proliferation in three-dimensional collagen were blocked by MMP inhibition, indicating that ROCK signaling enables PDAC cells to overcome extracellular matrix imposed restraints on invasion and growth. Treatment of mice with *Kras*[G12D]-driven PDAC with the ROCK small molecule inhibitor Fasudil prolonged survival. These findings reveal an ancillary role for increased ROCK signaling in advanced pancreatic cancer to promote extracellular matrix remodeling that enables invasive tumor growth by overcoming microenvironmentally imposed proliferation restraints. An implication of these results is that ROCK inhibitor administration to pancreatic cancer patients might reverse the ability of pancreatic cancer cells to surmount the growth-restraining properties of tumor-associated desmoplasia.

# Results

## ROCK2 expression increases during pancreatic cancer progression

To characterize ROCK2 expression in human pancreatic cancer progression, a tissue microarray containing 78 cases of pancreatic cancer and five normal pancreatic tissue samples was immunohistochemically stained with a ROCK2 antibody (Fig 1A). The specificity of the ROCK2 antibody had been validated in *Pdx1-Cre; ROCK2*[fl/fl]; *Rosa26-LSL-RFP* mouse pancreatic tissue, in which Cre-mediated recombination induced ROCK2 deletion and concomitant red fluorescent protein (RFP) expression (Appendix Fig S1A and B). ROCK2 levels rose with increasing tumor stage and grade, with the most progressed Stage III/IV having significantly higher levels than Normal or Stage I (Figs 1A and EV1A). In addition, ROCK2 was also present in tumor cells at resection margins (Fig EV1B). Analysis of The Cancer Genome Atlas (TCGA) Research Network (Cerami *et al*, 2012) pancreatic adenocarcinoma provisional dataset revealed that survival of patients with genomic amplification or significantly elevated mRNA for *ROCK1* and/or *ROCK2*, or truncating *ROCK1* mutations similar to previously described cancer-associated activating *ROCK1* truncation mutations (Lochhead *et al*, 2010), was significantly shorter than in patients without *ROCK1*/*ROCK2* alterations

**Figure 1.  ROCK expression is elevated in pancreatic cancer and promotes disease progression.**

A   ROCK2 immunohistochemistry-stained sections of normal and stages I, II, and III cases of human pancreas adenocarcinoma (left). Scale bar = 50 μm. Histoscores of ROCK2 staining (right) in normal pancreas (*n* = 5) and pancreas adenocarcinoma stage I (*n* = 22), stage II (*n* = 46), and stage III/IV (*n* = 4). One-way ANOVA with multiplicity adjusted exact *P*-value by *post hoc* Tukey multiple comparison test.
B   Overall survival of 107 patients without alterations versus 37 patients with ROCK1 and/or ROCK2 gene amplification significantly increased mRNA or truncation mutation from TCGA research network. Survival *P*-value determined by log-rank test.
C, D   Log2 median-centered ROCK1 or ROCK2 RNA expression in normal (*n* = 39) vs. PDAC (*n* = 39), normal (*n* = 5) vs. pancreatic adenocarcinoma (PAC) (*n* = 12), or normal (*n* = 6) vs. pancreatic carcinoma (PC) (*n* = 11) samples from indicated studies. Exact *P*-values determined by Mann–Whitney test.
E   ROCK1 and ROCK2 mRNA expression in human PDAC (*n* = 146) from the TCGA research network. Significance (*P*-value) of slope deviation from 0 determined by Deming regression.
F   Representative hematoxylin and eosin (H&E)- and ROCK2 immunohistochemistry-stained sections of normal mouse pancreas, acinar-to-ductal metaplasia (ADM), pancreatic intraepithelial neoplasia (PanIN) stages 1–3, and PDAC from mice with the indicated genotype. Scale bar = 50 μm.
G   Quantification of ROCK2 staining in pancreatic cells of normal (wildtype), ADM/PanIN1 (KC), PanIN2 (KPC), PanIN3 (KPC), and PDAC (KPC) tissue (*n* = 5 mice per group). One-way ANOVA with multiplicity adjusted exact *P*-value by *post hoc* Tukey multiple comparison test.
H   Survival analysis of *Pdx1-Cre; LSL-KRas*[G12D/+]; *LSL-Trp53*[R172H/+]; *LSL-ROCK2:ER* (RKPC) mice without (*n* = 21) or with conditional ROCK activation with tamoxifen citrate (*n* = 19). Survival *P*-value determined by log-rank test.

Data information: Box (upper and lower quartiles divided by median value) and whisker (5th–95th percentile) plots show outliers as individual points.

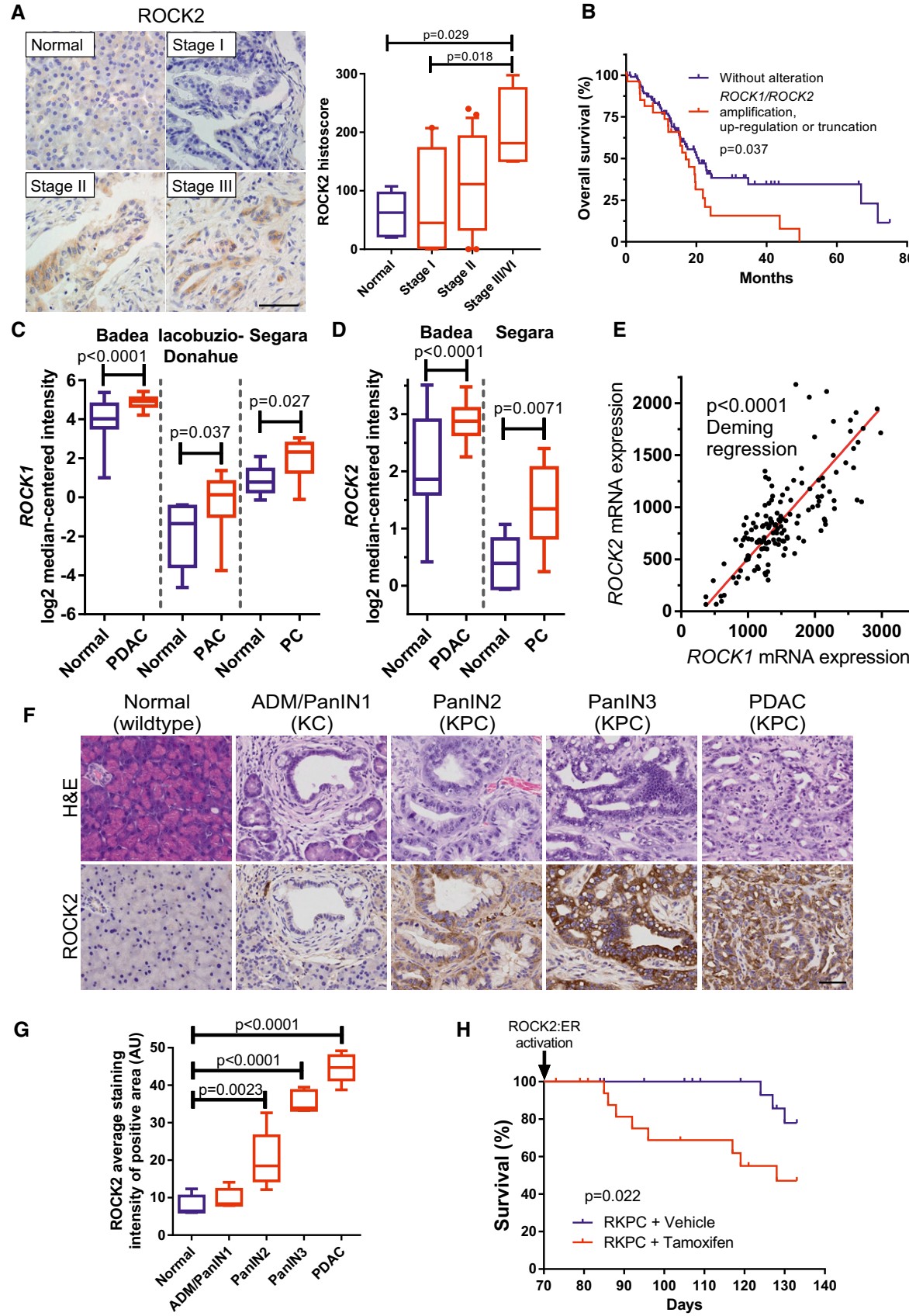

Figure 1.

(median survival periods: 21 months for patients without alterations versus 17 months, $P = 0.037$) (Fig 1B). These observations paralleled the significantly elevated *ROCK1* (Fig 1C) and *ROCK2* (Fig 1D) mRNA levels observed in pancreatic tumors relative to normal tissue detected in publicly available datasets (Iacobuzio-Donahue *et al*, 2003; Segara *et al*, 2005; Badea *et al*, 2008) using Oncomine (Rhodes *et al*, 2004). Furthermore, TCGA Research Network data (Cerami *et al*, 2012) also revealed significantly coordinated *ROCK1* and *ROCK2* mRNA expression in pancreatic cancers (Fig 1E), consistent with an observed advantage associated with increased ROCK signaling in pancreatic cancer (Laklai *et al*, 2016).

To further investigate ROCK2 expression in PDAC development and progression, genetically modified mouse models were used that closely recapitulate human PDAC (Gopinathan *et al*, 2015). Oncogenic *Kras*$^{G12D}$, preceded by a Lox-Stop-Lox (LSL) transcriptional termination cassette that can be excised by Cre recombinase expressed from the pancreas-selective *Pdx1* promoter (*Pdx1-Cre*), was combined with p53 deletion or mutation (Hingorani *et al*, 2003, 2005; Morton *et al*, 2010). ROCK2 immunostaining of normal pancreata as well as pancreatic tumors from *LSL-Kras*$^{G12D}$; *Pdx1-Cre* (KC) or *LSL-Kras*$^{G12D}$; *LSL-p53*$^{R172H}$; *Pdx1-Cre* (KPC) mice revealed weak ROCK2 expression in healthy normal tissue from wild-type mice, with progressive elevation in developing lesions of acinar-ductal metaplasia (ADM), pancreatic intraepithelial neoplasia (PanIN) stages 1–3, and highest expression in PDAC tumors (Fig 1F and G).

To determine whether increased ROCK signaling was sufficient to influence PDAC mouse survival, *LSL-Kras*$^{G12D}$; *LSL-p53*$^{R172H}$; *Pdx1-Cre* (KPC) mice were crossed with genetically modified mice containing a *Hprt*-targeted *LSL-ROCK2:ER* transgene (Fig EV2A) (Samuel *et al*, 2016) to establish a RKPC mouse line (Fig EV2B). Fusion of the ROCK2 kinase domain with the estrogen receptor (ER) hormone-binding domain generates a ROCK2:ER chimeric protein that is inactive in the absence of ligand, but which can be conditionally activated by estrogen analogues such as 4-hydroxytamoxifen (4HT) or tamoxifen (Fig EV2C) both *in vitro* and *in vivo* (Croft *et al*, 2006; Samuel *et al*, 2009, 2011; Sanz-Moreno *et al*, 2011; Kumar *et al*, 2012). To conditionally activate ROCK2:ER in *Pdx1*-expressing

*Kras*$^{G12D}$/*p53*$^{R172H}$ pancreatic tumor cells (Appendix Figs S2 and S3), tamoxifen citrate or vehicle control was administered to RKPC cohorts for 9 weeks, starting at 10 weeks of age when KPC mice have typically progressed to the PanIN stage (Fig 1F) (Morton *et al*, 2010). Conditional ROCK activation significantly ($P = 0.022$) reduced RKPC mouse survival time relative to vehicle control-treated mice (Fig 1H), indicating that the additional ROCK activity in the PDAC mouse model paralleled the effect of increased ROCK1/ROCK2 expression on reduced human PDAC patient survival (Fig 1B).

## ROCK kinases drive PDAC cell invasion and proliferation

Pancreatic tumor masses are largely composed of stroma, of which ECM proteins including collagen are major constituents (Feig *et al*, 2012). Tumor cells must invade through the collagen-rich microenvironment to invade local tissue and to metastasize. A three-dimensional collagen matrix invasion assay, used previously to characterize PDAC cell invasive behavior (Timpson *et al*, 2011; Nobis *et al*, 2014), in which tumor cells move through a fibroblast-remodeled collagen meshwork toward serum-containing medium, was utilized to determine how ROCK signaling might influence invasion. Invasive KPC mouse PDAC cells (Appendix Fig S4) (Timpson *et al*, 2011) invaded significantly less into collagen matrix in the presence of ROCK-selective inhibitor H1152 (Fig 2A).

To understand how additional ROCK signaling input might promote PDAC progression and reduce survival in human patients and mouse PDAC models (Fig 1), we adopted a conditional gain-of-function approach. Conditionally activated ER-fusions with ROCK1 (ROCK1:ER) or ROCK2 (ROCK2:ER) kinase domains, or green fluorescent protein (GFP:ER), were stably retrovirally transduced and expressed in non-invasive PDAC tumor cells derived from *Pdx1-Cre; LSL-Kras*$^{G12D}$; *LSL-p53*$^{fl/+}$ (KPflC) mice (Fig 2B). While antibodies specific for epitopes in the carboxyl-terminal regions present in full-length endogenous ROCK1 or ROCK2 showed consistent levels, blotting with an antibody against a kinase-domain epitope shared by endogenous ROCK1, ROCK2, ROCK1:ER, or ROCK2:ER revealed comparable expression of the fusion proteins to endogenous ROCK

---

**Figure 2.   ROCK activation induces PDAC cell invasion.**

A    H&E-stained sections of cell invasion into collagen matrix after 8 days. Scale bar = 100 μm. Invasion index of KPC cells in the presence or absence of 10 μM H1152 ROCK inhibitor. Means ± SEM (*n* = 9 for untreated, *n* = 8 for H1152), *P*-value by unpaired *t*-test.

B    Schematic representation of ROCK domains (RBD, Rho binding domain; PH, pleckstrin homology domain; CR, cysteine-rich). Conditionally activated human ROCK1, human ROCK2, and GFP control fusion proteins (EGFP, enhanced green fluorescent protein; hbER, estrogen receptor hormone-binding domain) were expressed in KPflC mouse PDAC cells and blotted with anti-GFP antibody.

C    KPflC cells expressing GFP:ER, ROCK1:ER or ROCK2:ER fusion proteins were treated for 24 h with EtOH vehicle or 1 μM 4HT in the presence or absence of 1 μM or 10 μM H1152. Immunoblotting shows endogenous ROCK1 and ROCK2, ER-fusions, and phosphorylation of MLC2 (T18S19). Total MLC (MRCL3/MRCL2/MYL9) and glyceraldehyde-3-phosphate dehydrogenase (GAPDH) were blotted as loading controls.

D    H&E-stained sections of cell invasion into collagen matrix after 8 days. Scale bar = 100 μm. Invasion index of KPflC cells treated with 1 μM 4HT. Means ± SEM (*n* = 6), one-way ANOVA with multiplicity adjusted exact *P*-value by *post hoc* Dunnett's multiple comparison test.

E    Cell proliferation determined by Ki67 immunofluorescence. Scale bar = 20 μm.

F    Quantification of cell number at the collagen matrix surface per 0.046 mm$^2$ field. Means ± SEM (*n* = 30), one-way ANOVA with multiplicity adjusted exact *P*-value by *post hoc* Dunnett's multiple comparison test.

G    Ki67-positive cell percentages at the surface and within the collagen matrix. Means ± SEM (*n* = 30; *n* = 12 for GFP:ER/Matrix), one-way ANOVA with multiplicity adjusted exact *P*-value by *post hoc* Dunnett's multiple comparison test.

H, I    Viable cell relative to starting cell numbers were determined after 24 or 48 h treatment with vehicle (−) or 4HT on uncoated plastic surfaces (H) or collagen1-coated surfaces (I). Means ± SEM (*n* = 3).

Source data are available online for this figure.

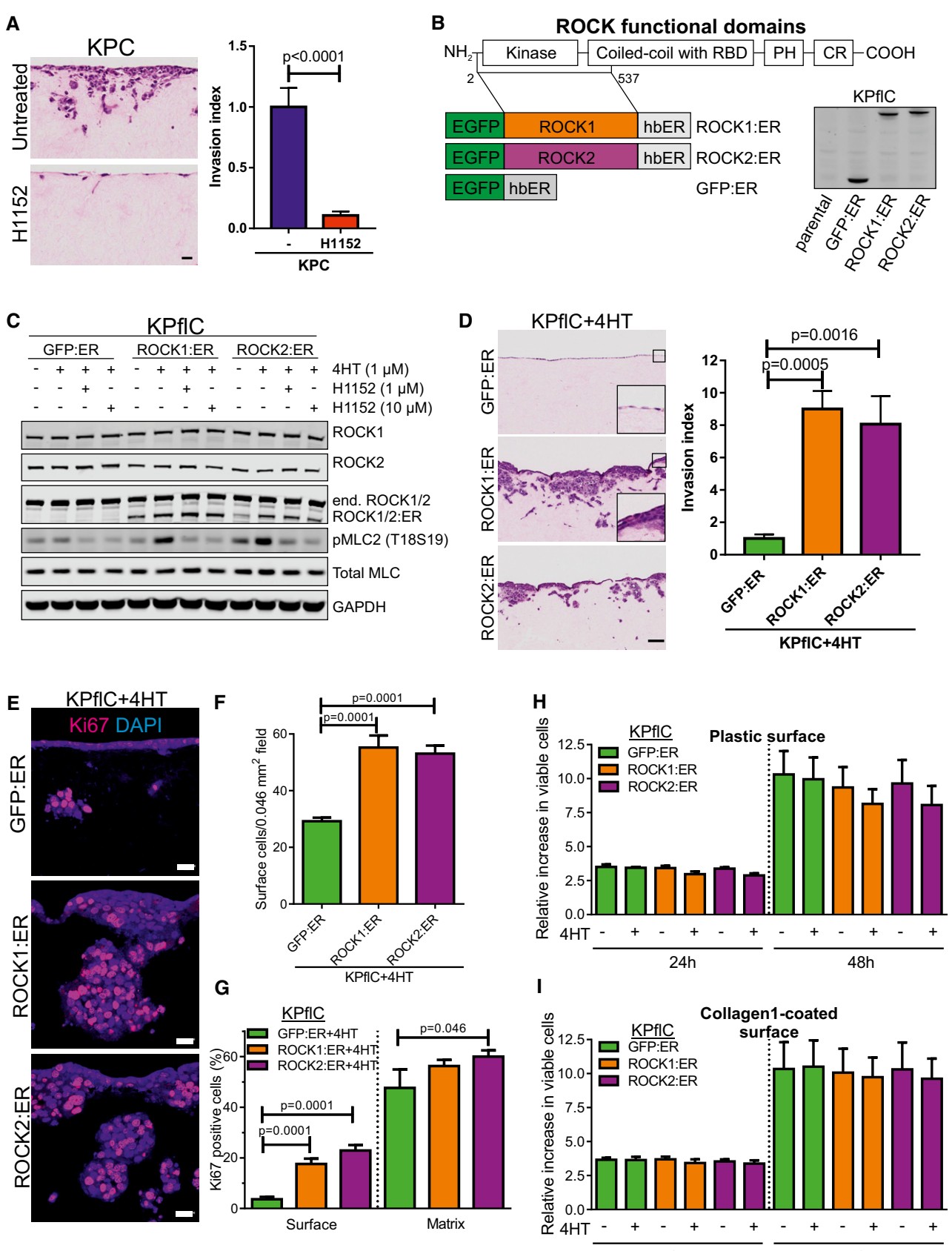

**Figure 2.**

kinases (Fig 2C). ROCK:ER fusions were induced and activated by 4HT, with consequent increased phosphorylation of MLC2 (pMLC2) that could be reversed by the ROCK-selective inhibitor H1152, in contrast to the lack of effect of 4HT treatment on GFP:ER-expressing cells (Fig 2C and Appendix Fig S5A). Consistent with previous observations (Croft *et al*, 2005), 4HT treatment of ROCK1:ER- or ROCK2:ER-expressing KPflC cells was sufficient to induce contraction and rounding compared to vehicle-treated controls, which were not observed for 4HT-treated GFP:ER-expressing cells (Appendix Fig S5B).

Non-invasive KPflC mouse PDAC cells (Appendix Fig S4; Timpson *et al*, 2011) expressing GFP:ER and treated with 4HT did not invade the collagen matrix, while ROCK1:ER- or ROCK2:ER-expressing cells were significantly more invasive (Fig 2D). Comparable results were obtained if embedded fibroblasts were not removed prior to addition of ROCK1:ER- or ROCK2:ER-expressing KPflC cells to the collagen matrix surface (Fig EV3A). Further analysis revealed thicker ROCK1:ER and ROCK2:ER cell layers at collagen matrix surfaces (Fig 2D, inserts) with significantly more cells relative to GFP:ER-expressing cells (Fig 2E and F), associated with significantly increased cell proliferation as indicated by Ki67 immunoreactivity (Fig 2E and G). All three cell lines had high Ki67-positive percentages of cells that had successfully invaded collagen matrix, although both ROCK:ER-expressing cell lines tended to have higher percentages than GFP:ER controls (Fig 2G). However, when cells were grown on two-dimensional plastic (Fig 2H) or a thin collagen1-coated surface (Fig 2I), conditional ROCK activation did not affect viable proliferation. Furthermore, the invasion and proliferation of ROCK1:ER or ROCK2:ER-expressing cells were significantly reduced by addition of H1152 ROCK inhibitor to the collagen matrix invasion assay (Fig EV3B–E). Together, these data indicate that increased ROCK activity drives PDAC cell invasion, which enables proliferation by overcoming restraints imposed by three-dimensional collagen matrices that are not influential in two dimensions.

## ROCK activation induces collagen remodeling

To determine whether ROCK-induced invasion (Fig 2) was associated with collagenolysis, collagen matrix sections were immunostained with Col1 3/4C antibody to detect a collagen1 neo-epitope at the C-terminal end of the ¾ fragment resulting from α1- and α2-chains cleavage at G775–I776 and G775–L776, respectively, and local triple helix denaturation. While collagen cleavage was minimal in GFP:ER-expressing cells, invasive ROCK1:ER and ROCK2:ER cells had considerable staining below collagen matrix surfaces and surrounding cell clusters (Fig 3A, left). To quantitatively assess fibrillar collagen organization in collagen matrices, second harmonic generation (SHG) microscopy, which takes advantage of optical

properties of supramolecularly assembled collagen fibers (Chen *et al*, 2012), was performed at multiple *z*-planes below the collagen matrix surface (Fig 3A, second to fourth panels). When the intensity of fibrillary collagen detected by SHG was assessed, it was apparent that fibrillar collagen areas diminished more slowly with increasing depth of surface penetration in GFP:ER-expressing cells (Fig 3A, green line) compared to the greater decline observed in ROCK1:ER-expressing cells (Fig 3A, orange line). Gray-level correlation matrix (GLCM) texture analysis (Mostaço-Guidolin *et al*, 2013; Cameron *et al*, 2015; Miller *et al*, 2015) of SHG images confirmed that invading ROCK:ER cells had extensively remodeled fibrillar collagen (Fig 3B and C). Transmission electron microscopy (TEM) revealed that regions of collagen matrix surfaces not associated with invasive cell clusters lacked protrusions (Fig 3D), while surface cells near invasive clusters (Fig 3D, black border) or invading cell clusters (Fig 3D, red or green borders) had extensive protrusions resembling blebs and microvesicles. Tannic acid-glutaraldehyde fixation (Cotta-Pereira *et al*, 1976) allowed visualization of collagen fibers within the collagen matrix. Long fibers were often visible in regions away from cells or adjacent to non-invading cells at the collagen matrix surface (Fig 3E; left, yellow arrows), while there were decreased apparent long fibers and shorter collagen bundles observed adjacent to invading cell clusters, protrusions and microvesicles (Fig 3E; right, red arrows). These results reveal that ROCK:ER-expressing cells extensively re-modeled three-dimensional collagen matrices, likely resulting from proximal collagenolysis, physical force from actomyosin contraction, and pressure produced by invasive cell cluster expansion.

## ROCK regulation of gene expression and MMP release

Although ROCK actions on actin structures and cell morphology are well-known (Rath & Olson, 2012; Julian & Olson, 2014), the same signaling pathway regulates gene transcription (Rajakylä & Vartiainen, 2014). Using RNA sequencing, expressions of 305 genes were found to be significantly ($P < 0.05$) changed greater than twofold in 4HT-treated ROCK1:ER-expressing cells relative to GFP:ER-expressing cells (Fig 4A, left), while 374 genes were significantly changed in ROCK2:ER versus GFP:ER-expressing cells (Fig 4A, right), of which 285 were common changes (Fig 4B) (Rath *et al*, 2016). The greatest significantly increased fold-changes were *Mmp10* and *Mmp13*, prostaglandin-endoperoxidase synthase 2 (*Ptgs2* also known as *Cox2*), and Tenascin-C (*Tnc*) (Fig 4A and Dataset EV1). *COX2* is commonly elevated in human PDAC (Yip-Schneider *et al*, 2000), and its overexpression promoted mouse PDAC development (Hill *et al*, 2012). TNC is a component of tumor-specific ECM associated with pancreatic carcinogenesis (Esposito *et al*, 2006). Metacore™ was used to

**Figure 3. ROCK activation induces collagen cleavage and remodeling.**

A    Immunofluorescence of collagen matrix sections co-stained for cleaved collagen neo-epitope Col1 3/4C and DAPI (left panels). Second harmonic generation (SHG) images of collagen fibers within three-dimensional collagen matrices at 15 µm and 45 µm below the cell surface (middle panels). Scale bar = 20 µm. Graphs indicate the mean ± SEM fibrillary collagen areas at 2.5 µm increments below the cell/collagen interface (*n* = 5). SHG fibrillar collagen (red); cells (green).

B, C    Gray-level correlation matrix (GLCM) texture analysis of SHG correlation (*n* = 6) and contrast (*n* = 5). Mean ± SEM at each distance.

D    Transmission electron microscopy of a collagen matrix section. Scale bar = 1 µm. Positions of magnified areas indicated by corresponding colored squares.

E    Transmission electron microscopy of collagen matrix sections with collagen stained by Tannic acid. Yellow arrows indicate long collagen bundles. Red arrows indicate short collagen bundles. Scale bar = 0.5 µm.

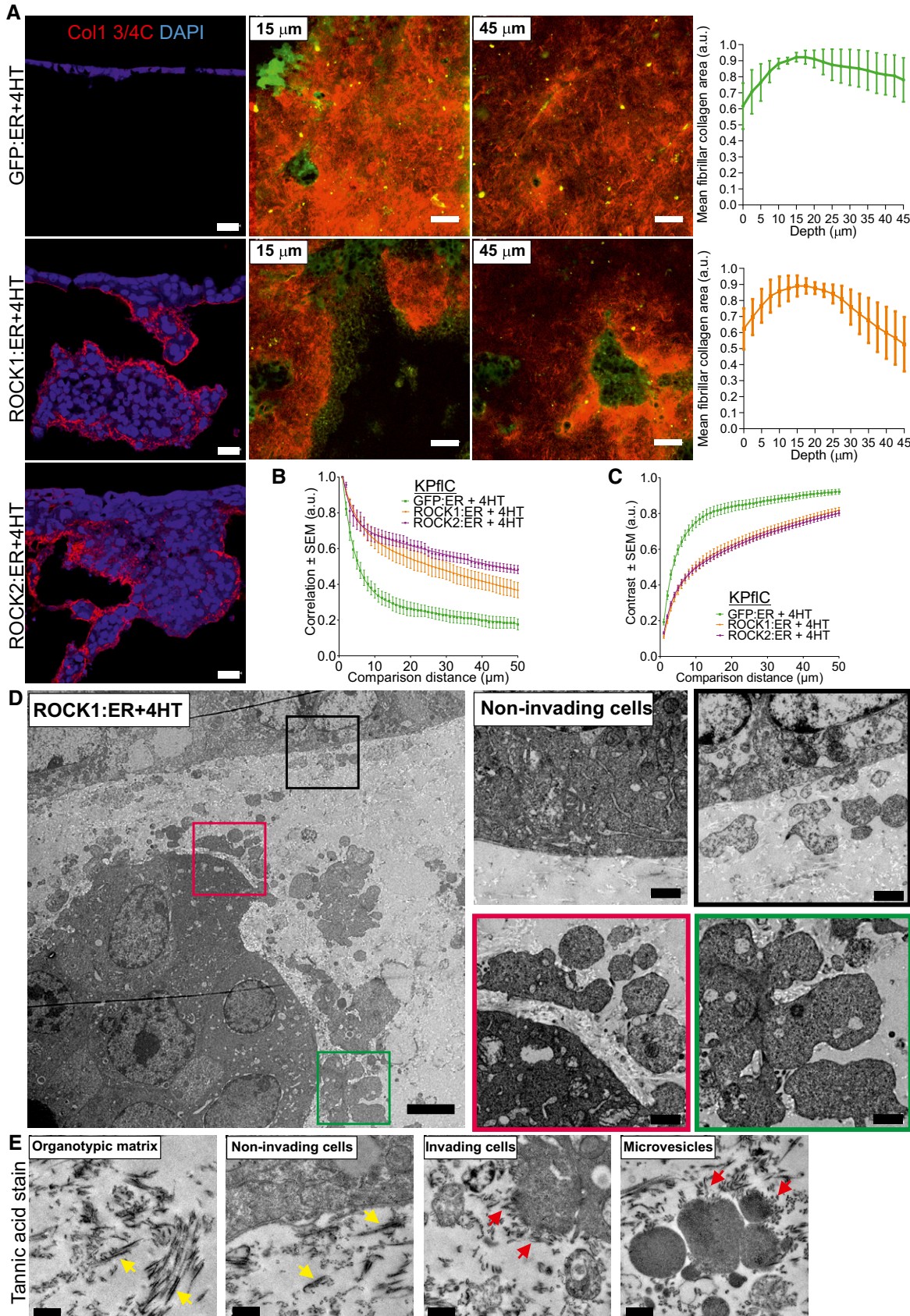

**Figure 3.**

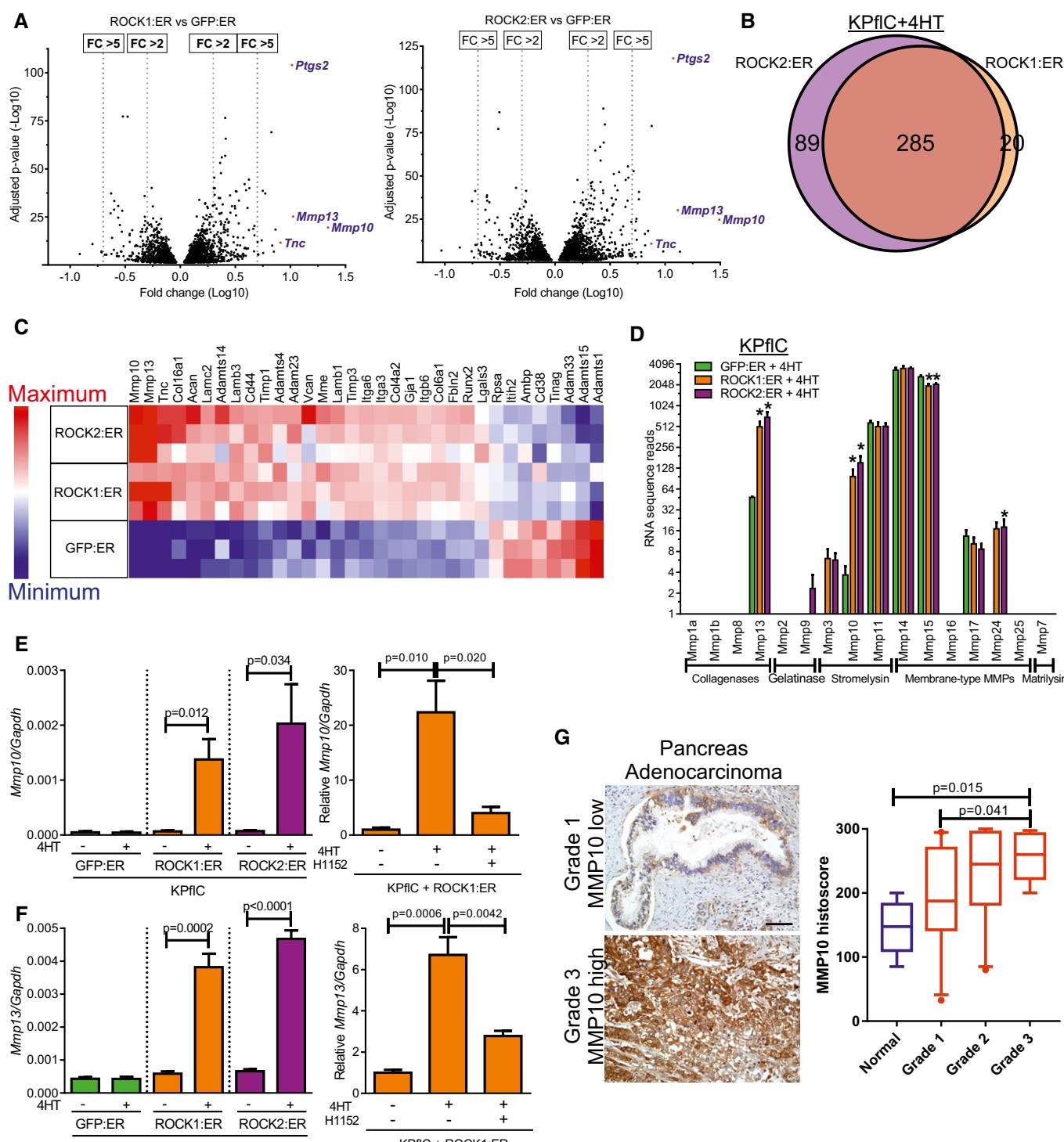

**Figure 4.**

identify differentially expressed gene process networks (1,036 genes with false discovery rates < 5% and ≥ 1.5-fold increased (603) or decreased (433) in ROCK1:ER + 4HT vs. GFP:ER + 4HT and ROCK2:ER + 4HT vs. GFP:ER + 4HT), which ranked *Cell adhesion: Cell-matrix interaction* first (Appendix Table S1). Figure 4C shows the relative expression of 33 differentially expressed genes from this network in 4HT-treated GFP:ER-,

ROCK1:ER-, or ROCK2:ER-expressing cells from three independent experiments. The average number of MMP sequencing reads from 4HT-treated GFP:ER-, ROCK1:ER-, and ROCK2:ER-expressing cells were plotted in Fig 4D, indicating the significant up-regulation of the stromelysin *Mmp10* and collagenase *Mmp13* (Appendix Table S2). Although not a direct collagenase, MMP10 superactivates procollagenases including proMMP13 (Barksby

**Figure 4.  MMP induction by ROCK activation.**

A   Volcano plots of RNA sequencing data of genes that pass a threshold false discovery rate (FDR) < 5%, with log10 fold-change (FC) in expression in 4HT-treated ROCK1:ER-expressing cells relative to GFP:ER-expressing cells (left) or 4HT-treated ROCK2:ER-expressing cells relative to GFP:ER-expressing cells (right) versus log10-adjusted $P$-value ($n = 3$). ROCK1:ER + 4HT vs GFP:ER + 4HT: 3,828 genes. ROCK2:ER + 4HT vs. GFP:ER + 4HT: 4,481 genes.

B   Venn diagram of genes with FDR < 5% and ≥ 2-fold expression changes for ROCK1:ER + 4HT vs. GFP:ER + 4HT and ROCK2:ER + 4HT vs. GFP:ER + 4HT.

C   Heatmap of 33 differentially expressed genes in the MetaCore™ process network Cell adhesion: Cell-matrix interaction. High relative expression in red and low relative expression in blue.

D   MMP average RNA sequence reads. Mean ± SEM ($n = 3$). * indicates adjusted $P$-value < 0.05.

E   Left: *Mmp10* mRNA levels relative to *Gapdh* determined by qPCR following treatment with EtOH vehicle (−) or 1 μM 4HT for 24 h. Means ± SEM ($n = 4$), $P$-value by unpaired $t$-test. Right: *Mmp10* mRNA levels relative to *Gapdh* determined by qPCR following treatment with EtOH vehicle (−), 1 μM 4HT or 4HT + 1 μM H1152 ROCK inhibitor for 24 h. Means ± SEM ($n = 3$), one-way ANOVA with multiplicity adjusted exact $P$-value by *post hoc* Tukey multiple comparison test.

F   Left: *Mmp13* mRNA levels relative to *Gapdh* determined by qPCR following treatment with EtOH vehicle (−) or 1 μM 4HT for 24 h. Means ± SEM ($n = 4$), $P$-value by unpaired $t$-test. Right: *Mmp13* mRNA levels relative to *Gapdh* determined by qPCR following treatment with EtOH vehicle (−), 1 μM 4HT or 4HT + 1 μM H1152 ROCK inhibitor for 24 h. Means ± SEM ($n = 3$), one-way ANOVA with multiplicity adjusted exact $P$-value by *post hoc* Tukey multiple comparison test.

G   MMP10 immunohistochemistry-stained sections of human pancreas adenocarcinoma (left). Scale bar = 100 μm. Histoscores of MMP10 staining (right) in normal pancreas ($n = 5$) and pancreas adenocarcinoma grade 1 ($n = 23$), grade 2 ($n = 27$), and grade 3 ($n = 13$). One-way ANOVA with multiplicity adjusted exact $P$-value by *post hoc* Tukey multiple comparison test. Box (upper and lower quartiles divided by median value) and whisker (5th–95th percentile) plots show outliers as individual points.

*et al*, 2006). Expression of several collagens was also altered by ROCK activation (Appendix Fig S6 and Appendix Table S3). ROCK-induced increases in *Mmp10* (Fig 4E) and *Mmp13* (Fig 4F) mRNA transcripts were confirmed by qPCR. In addition, the ROCK inhibitor H1152 reversed ROCK-induced *Mmp10* and *Mmp13* expression (Fig 4E and F). Furthermore, higher MMP10 histoscores correlated with higher pancreatic adenocarcinoma grades (Fig 4G).

Immunofluorescence indicated that ROCK:ER activation led to MMP10 and MMP13 accumulation in bleblike protrusions (Fig 5A), effects that could be reversed by H1152 ROCK inhibitor (Fig EV4A and B). Conditioned media contained increased MMP10 and MMP13 following ROCK:ER activation, with little difference in MMP10 levels in cell lysates (Fig 5B), suggesting that MMP protein release was not rate-limiting. Given that MMP10 release has been reported to be mediated via microvesicles (de Lizarrondo *et al*, 2012) and that ROCK signaling was shown to promote microvesicle formation (Li *et al*, 2012), we sought to determine whether ROCK activation promoted MMP10 release via microvesicles. Consistent with the appearance of microvesicles associated with ROCK activation in collagen matrix invasion assays (Fig 3D and E), microvesicle enrichment by ultracentrifugation of conditioned media yielded significantly more protein following ROCK activation (Fig 5C). Western

blotting revealed significantly increased MMP10 protein, which could be detected in isolated microvesicles by immunogold staining in TEM images, as well as increased MMP10 to caveolin and MMP10 to microvesicle protein ratios in ultracentrifuge-enriched protein (Fig 5D). In contrast, the caveolin to microvesicle protein ratio (Fig 5E) as well as MMP10 to GAPDH or caveolin to GAPDH ratios in cell lysates (Fig 5F) were not affected by ROCK activation. Together, these findings indicate that ROCK-induced *Mmp10* and *Mmp13* mRNA expression was associated with increased release of microvesicles that enable efficient MMP release into the surrounding environment.

## ROCK kinases induce collagen remodeling to enable invasive growth

To directly test whether ROCK-induced MMP release drives collagen degradation, GFP:ER- and ROCK1:ER-expressing cells were plated on FITC-labeled collagen1 and collagenolysis was detected by reduced FITC fluorescence. Greater areas of collagen degradation were apparent in ROCK1-activated cells (Fig 6A and B), which were significantly reduced by H1152 to below GFP:ER control levels (Fig 6A and B). Treatment with the myosin ATPase inhibitor Blebbistatin to reduce actomyosin contraction

**Figure 5.  MMP proteins are efficiently released in response to ROCK activation.**

A   Confocal microscope images of ROCK1:ER-expressing cells co-stained for F-actin, MMP10 (left) or MMP13 (right), and DAPI following treatment with vehicle or 1 μM 4HT for 24 h. Multiple z-planes were used to generate $x$–$z$ and $y$–$z$ images. Scale bar = 5 μm.

B   Representative immunoblot of MMP13 and MMP10 in conditioned media, as well as MMP10 and GAPDH in cell lysates. Cells were treated with vehicle (−) or 1 μM 4HT for 48 h.

C   Representative stained gel of ultracentrifuge-enriched microvesicle protein from GFP:ER or ROCK1:ER-expressing cell-conditioned media following treatment with vehicle (−) or 1 μM 4HT for 24 h (top). Absolute arbitrary unit values for total microvesicle protein levels in stained gels (bottom). Means ± SEM ($n = 4$), $P$-value by ratio paired $t$-test.

D   Representative immunoblot of MMP10 and caveolin in ultracentrifuge-enriched microvesicle proteins from cell-conditioned media following treatment with vehicle (−) or 1 μM 4HT for 24 h as well as MMP10, caveolin, and GAPDH in cell lysates (top left). Transmission electron microscopy of ultracentrifuge-enriched microvesicles (top center), and immunogold labeling of MMP10 in ROCK1:ER 4HT-treated microvesicles, indicated by red arrows (top right). Absolute arbitrary unit values for MMP10 levels in Western blots of ultracentrifugation-enriched microvesicle proteins (bottom left). Ratios of MMP10 to caveolin (bottom center) or to total microvesicle proteins (bottom right). Means ± SEM ($n = 4$), $P$-value by ratio paired $t$-test.

E   Absolute arbitrary unit values for caveolin levels in Western blots of ultracentrifugation-enriched microvesicle proteins (left). Ratio of caveolin to total microvesicle proteins (right). Means ± SEM ($n = 4$), $P$-value by ratio paired $t$-test.

F   Ratios of MMP10 to GAPDH (left) and caveolin to GAPDH (right) in Western blots of whole-cell lysates. Means ± SEM ($n = 3$), $P$-value by ratio paired $t$-test.

Source data are available online for this figure.

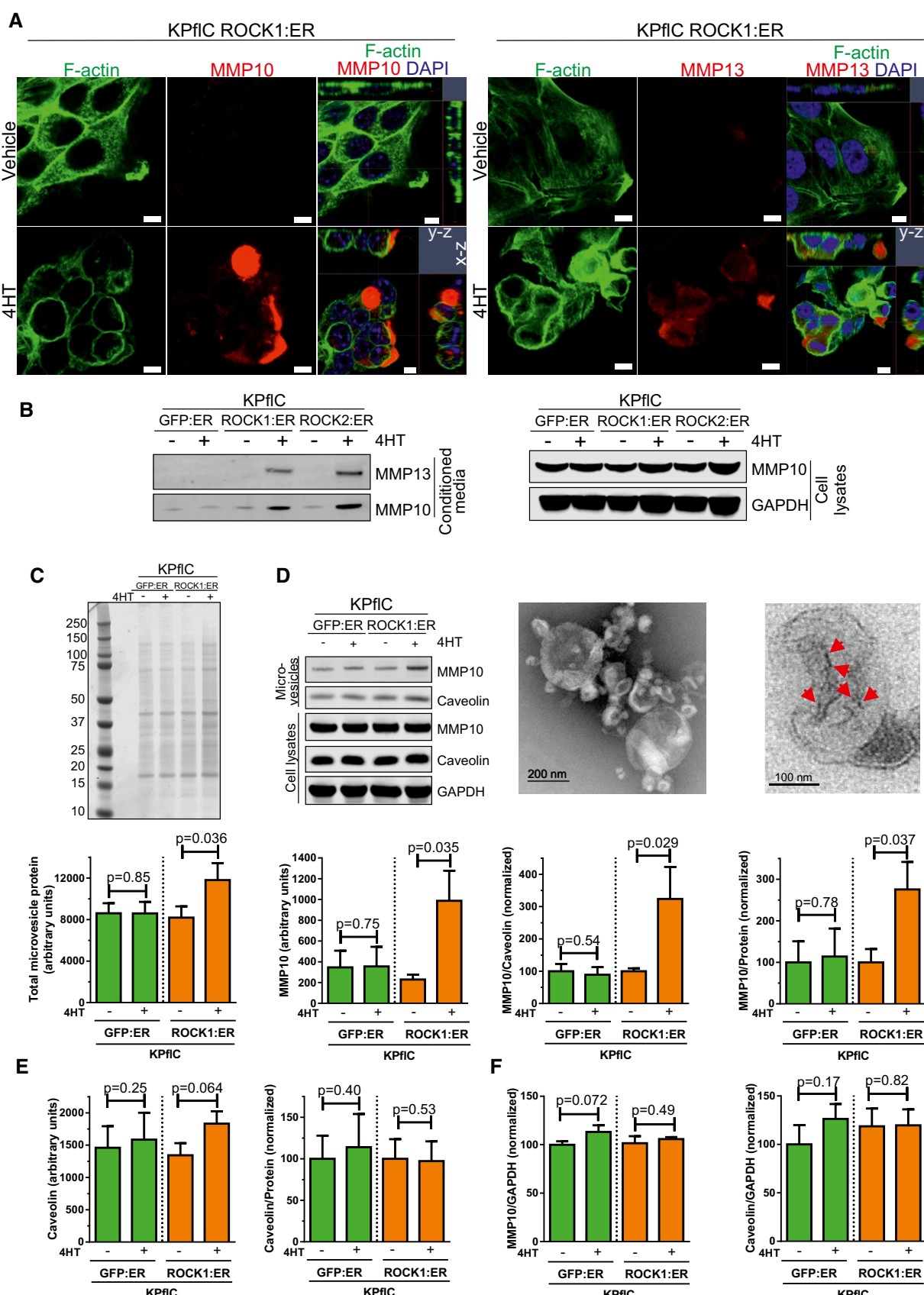

**Figure 5.**

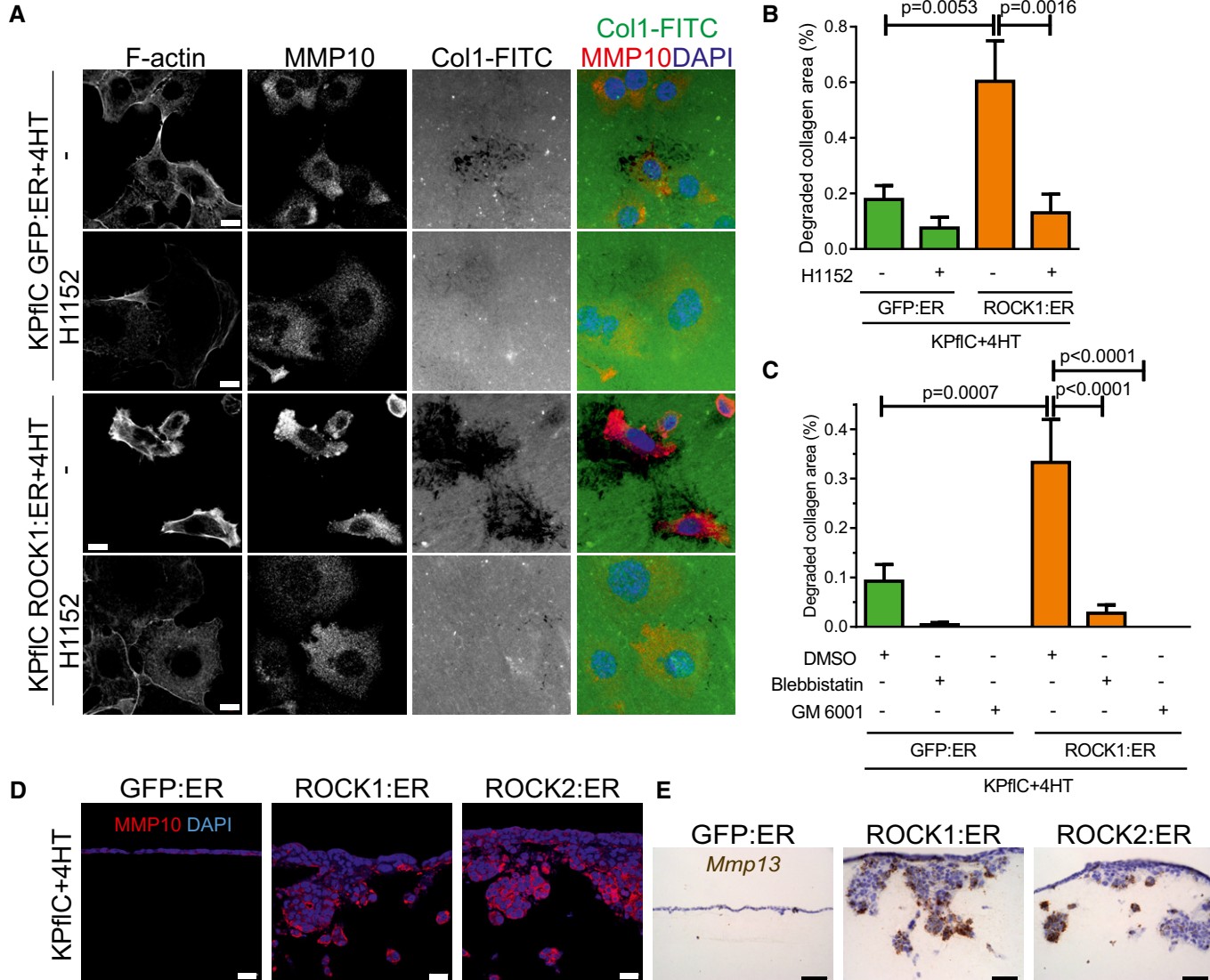

**Figure 6.  ROCK activation promotes collagen degradation.**

A   Representative fluorescence images of cells treated with 1 μM 4HT in the absence or presence of 10 μM H1152 for 18 h on Collagen1-FITC (Col1-FITC). Co-staining for F-actin, MMP10 (red), and DAPI (blue). Scale bar = 10 μm.

B   Quantification of collagen degradation by cells treated with 1 μM 4HT or 4HT + 10 μM H1152 for 16 h. Means ± SEM (*n* = 15), one-way ANOVA with multiplicity adjusted exact *P*-value by *post hoc* Tukey multiple comparison test.

C   Quantification of collagen degradation by cells treated with 1 μM 4HT, 4HT + DMSO vehicle, 4HT + 50 μM Blebbistatin or 4HT + 10 μM GM6001 for 16 h. Means ± SEM (*n* = 15), one-way ANOVA with multiplicity adjusted exact *P*-value by *post hoc* Tukey multiple comparison test.

D   Immunofluorescence of collagen matrix sections co-stained for MMP10 and DAPI. Scale bar = 20 μm.

E   *Mmp13 in situ* hybridization-stained sections of collagen matrices. Scale bar = 50 μm.

(Straight *et al*, 2003) or the broad-spectrum MMP inhibitor GM6001 (Grobelny *et al*, 1992) reduced GFP:ER control and ROCK1:ER-induced collagen degradation (Fig 6C). Further, ROCK-induced MMP10 and *Mmp13* were present at invasive cell interfaces with collagen matrix (Fig 6D and E). The critical importance of MMP activity for ROCK-induced three-dimensional collagen matrix invasion was demonstrated by sensitivity to GM6001 (Fig 7A and B). While ROCK:ER activation increased the total number of cells (Fig 2F) and Ki67-positive cells at the collagen matrix surface (Fig 2G), GM6001 blocked ROCK-induced cell layer thickening (Fig 7A, inserts and C) and significantly reduced the number of Ki67-positive cells (Fig 7D and E). Although GM6001 significantly blocked ROCK-induced invasion (Fig 7A and B), the Ki67 positivity of successfully invaded cells was not affected by GM6001 (Fig 7D and F). Proliferation was not affected by GM6001 on two-dimensional uncoated (Fig 7G) or collagen1-coated surfaces (Fig 7H). These results indicate that ROCK activation enables PDAC proliferation by promoting ECM degradation to relieve constraints imposed by three-dimensional microenvironments.

## ROCK inhibition promotes PDAC mouse survival

Finally, we wished to determine whether ROCK signaling was a pharmacologically actionable target in the mouse pre-clinical PDAC model. The ROCK inhibitor Fasudil has been safely used in Japan since 1995 to reverse the effect of ROCK-mediated blood vessel constriction during cerebral vasospasm (Olson, 2008). Administration of Fasudil to KPC mice from 10 weeks of age significantly increased survival (Fig 8A), with 123-day median survival for vehicle-treated mice and 168-day median survival for Fasudil-treated mice ($P = 0.043$). Endpoint tumors showed many similar characteristics (Appendix Fig S7A–E), including the staining of collagen with picrosirius red (Lattouf *et al*, 2014) (Fig 8B) revealing no difference in total collagen area (Fig 8C, top). However, pancreatic tumors from Fasudil-treated mice had significantly higher average collagen staining intensity than vehicle controls over the positive staining regions (Fig 8C, bottom), indicating that ROCK inhibition was associated with increased collagen content. These results reveal an association between pancreatic cancer survival in mice with increased collagen in PDAC tumors following ROCK inhibition.

## Discussion

Previous studies have reported significantly higher levels of *ROCK1* (Iacobuzio-Donahue *et al*, 2003; Segara *et al*, 2005; Badea *et al*, 2008) (Fig 1C) and *ROCK2* (Segara *et al*, 2005; Badea *et al*, 2008) (Fig 1D) mRNA in pancreatic tumors relative to normal tissue. ROCK1 protein was also detected in 18 of 21 pancreatic cancer samples and five cell lines, but not in 10 normal pancreas specimens (Kaneko *et al*, 2002). Regression analysis of data from the TCGA Research Network (Cerami *et al*, 2012) revealed that *ROCK1* and *ROCK2* mRNA expression appear to be coordinately regulated (Fig 1E). In this study, using a rigorously validated anti-ROCK2 antibody (Appendix Fig S1A), we found that ROCK2 expression increased with tumor progression in human PDAC (Fig 1A) and in mouse pancreatic cancer models (Fig 1F and G). Mutant Kras induction of eukaryotic translation initiation factor 5A, which plays a critical role in PDAC tumor growth (Fujimura *et al*, 2014), facilitates elevation of ROCK1 and ROCK2 protein levels in pancreatic cancer cells (Fujimura *et al*, 2015), indicating that there are also post-transcriptional mechanisms that may contribute to increased ROCK expression in PDAC. These results suggest that, although ROCK1 and ROCK2 are unlikely to be cancer drivers, their frequent elevated expression in advanced tumors is consistent with their providing ancillary functions.

The role of the PDAC-associated desmoplastic reaction in tumor growth is unclear and controversial. Although the dense stroma limits drug efficacy due to poor tumor vascularization (Olive *et al*, 2009) and high interstitial fluid pressure (Provenzano *et al*, 2012), anti-stromal therapies have not been approved for clinical use (Neesse *et al*, 2015). Recent studies found that mice with reduced stroma developed more aggressive tumors with undifferentiated histology, increased proliferation, and reduced survival (Ozdemir *et al*, 2014; Rhim *et al*, 2014), suggesting that the desmoplastic microenvironment serves to restrain PDAC tumor growth. Consistent with these observations, high stromal collagen density has been associated with better pancreatic cancer patient prognosis (Bever *et al*, 2015). In opposition to these findings, high levels of large diameter fibrillary collagen adjacent to tumor margins have been associated with poor patient survival (Miller *et al*, 2015; Laklai *et al*, 2016). These apparently contradictory results suggest that PDAC desmoplasia may have both negative and positive properties that act in opposition. Thus, the extent of desmosplasia alone is not likely to correlate directly with tumor aggressiveness or patient outcome. By implication, PDAC cells are likely to favor properties, particularly in advanced tumors with dense stromal components, which enable them to overcome inhibitory constraints on tumor growth imposed by the desmoplastic microenvironment.

Matrix metalloproteinases are proteolytic enzymes that influence tissue homeostasis and tumor growth through processes including ECM remodeling and tissue invasion (Kessenbrock *et al*, 2010). It was recently shown that leptin, which uses ROCK as a critical signal mediator (Huang *et al*, 2012), induces migration and invasion of human pancreatic cells via MMP13 up-regulation, while increased MMP13 expression in patients was associated with lymph node metastasis and pathological stage (Fan *et al*, 2015). MMP10 was also found to enhance the proliferation, invasion, and metastatic potential of PDAC cells (Zhang *et al*, 2014). MMP10 lacks collagenase activity itself, but MMP10 was found to play a critical role in tissue remodeling by promoting the expression and/or activity of collagenolytic MMPs, in particular MMP13 (Barksby *et al*, 2006; Rohani *et al*, 2015).

Although expression of some collagens increased in mouse PDAC cells following ROCK activation (Appendix Fig S6 and Appendix Table S3), the most profound gene expression changes were in networks associated with extracellular matrix remodeling (Fig 4C and D, and Appendix Table S1). Recently, it was found that actomyosin contractile force in rounded melanoma cells increased the expression of several MMPs and that these MMPs promoted amoeboid migration (Orgaz *et al*, 2014). Consistent with these findings, our gene expression profiling of PDAC cells

---

**Figure 7. ROCK activation promotes MMP-dependent invasive growth.**

A H&E-stained sections of cell invasion into collagen matrix after 8 days, in the absence (top) or presence of 10 μM GM6001 (bottom). Scale bar = 100 μm.

B Invasion index of KPflC cells. Means ± SEM ($n = 4$), $P$-value by unpaired $t$-test.

C Quantification of cell number at the collagen matrix surface per 0.046 mm² field. Means ± SEM ($n = 20$), $P$-value by unpaired $t$-test.

D Cell proliferation determined by Ki67 immunofluorescence. Scale bar = 20 μm.

E Ki67-positive cell percentages at the collagen matrix surface. Means ± SEM ($n = 20$; $n = 19$ for GFP:ER/vehicle, $n = 18$ for GFP:ER/GM6001), one-way ANOVA with multiplicity adjusted exact $P$-value by *post hoc* Tukey multiple comparison test.

F Ki67-positive cell percentages in collagen matrix. Means ± SEM ($n = 20$; $n = 8$ for GFP:ER/vehicle, $n = 10$ for GFP:ER/GM6001, $n = 17$ for ROCK1:ER/GM6001, $n = 15$ for ROCK2:ER/GM6001), one-way ANOVA with multiplicity adjusted exact $P$-value by *post hoc* Tukey multiple comparison test.

G, H The increase in viable cell numbers of cells plated on uncoated plastic surfaces (G) or collagen1 (Col1)-coated surfaces (H) and treated with 1 μM 4HT for 24 h not affected by 10 μM GM6001. Means ± SEM ($n = 3$), one-way ANOVA with multiplicity adjusted exact $P$-value by *post hoc* Tukey multiple comparison test.

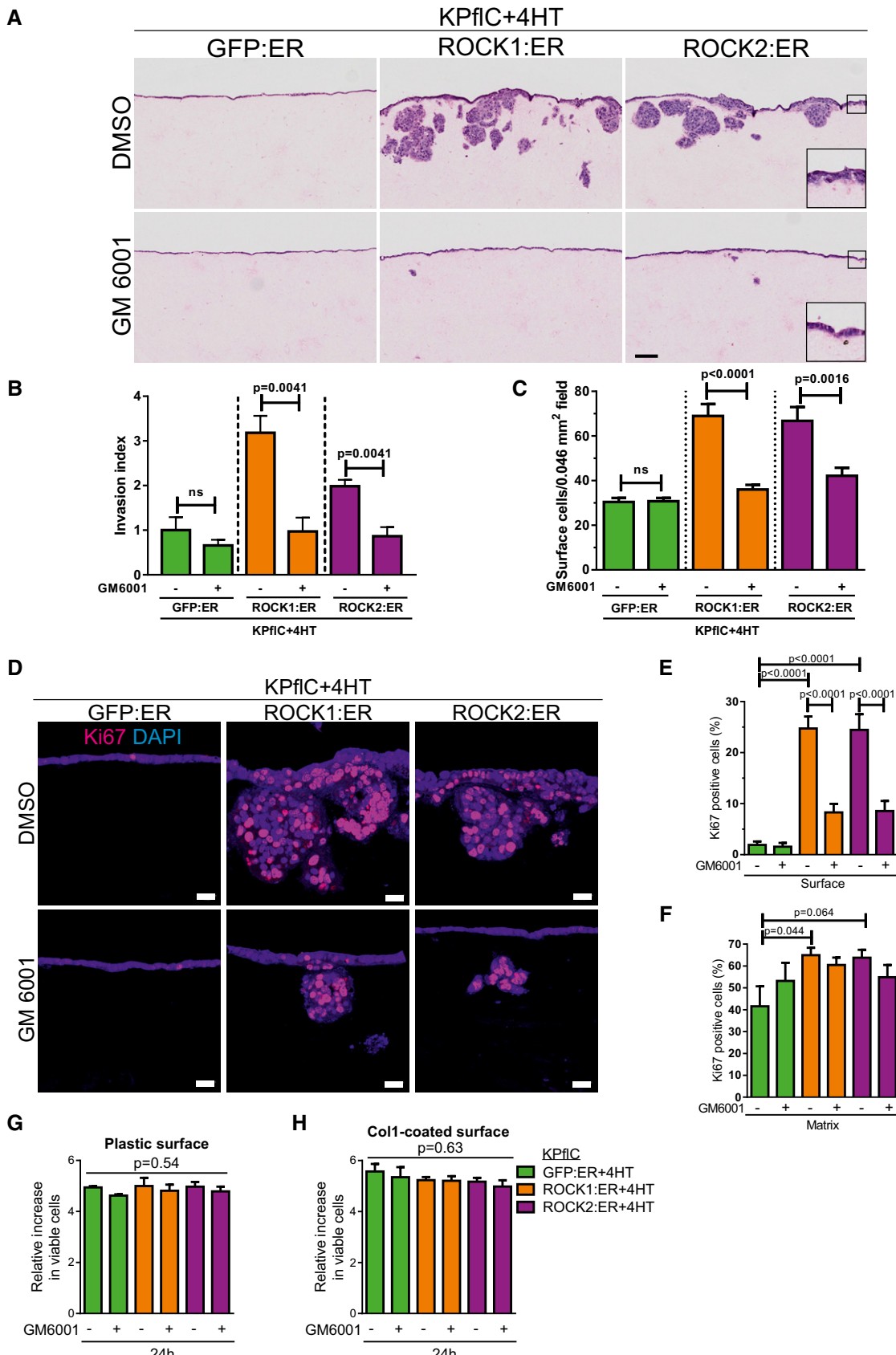

**Figure 7.**

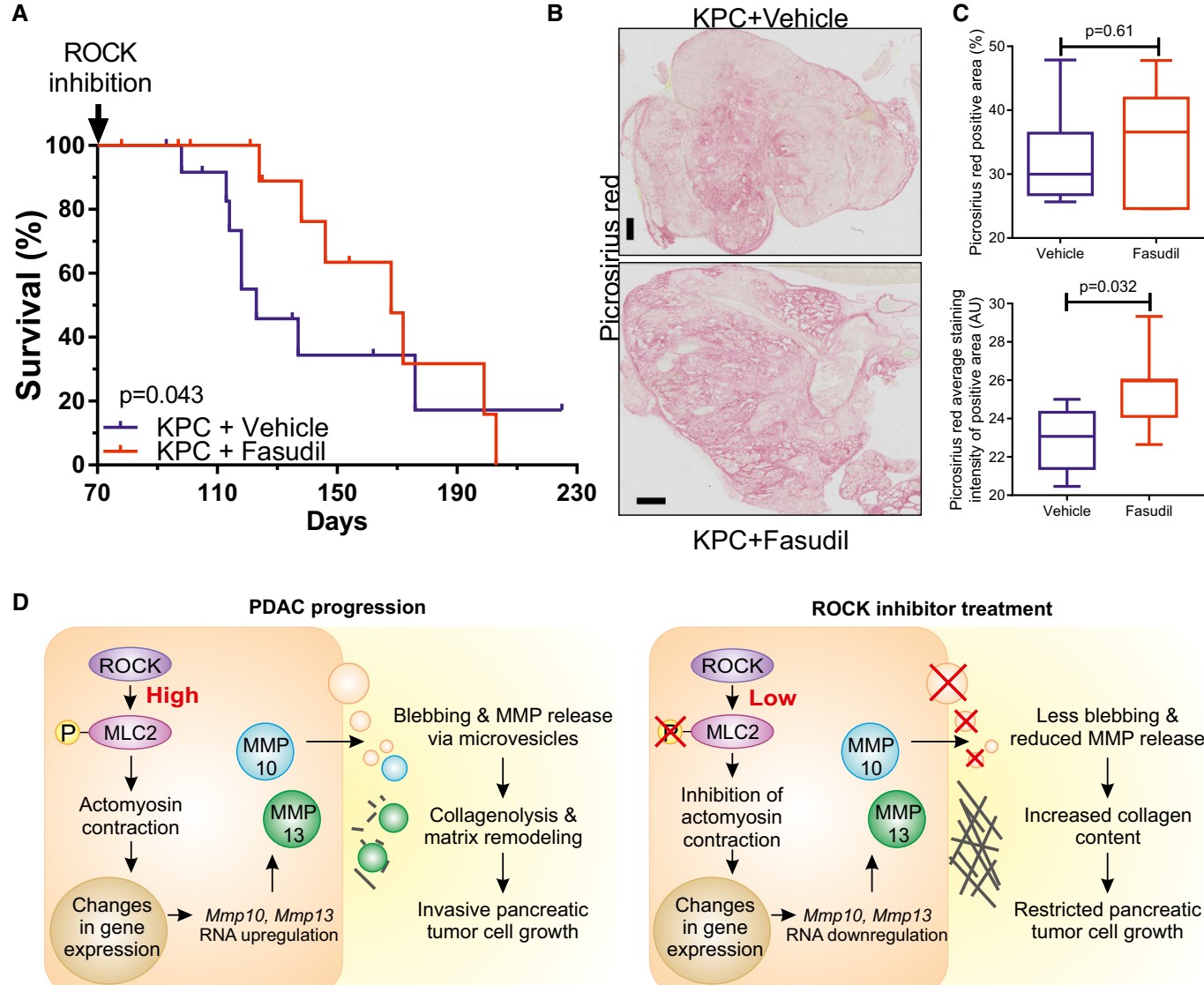

**Figure 8.  ROCK kinase signaling influences PDAC mouse survival.**

A   Survival analysis of *Pdx1-Cre; LSL-KRas^{G12D/+}; LSL-Trp53^{R172H/+}* (KPC) mice without (*n* = 13) or with Fasudil treatment (*n* = 13). Survival *P*-value determined by Gehan–Breslow–Wilcoxon test.

B   Representative images of picrosirius red stained endpoint tumors from KPC mice. Scale bar = 1 mm.

C   Picrosirius red percentage of positive staining area (top) or average staining intensity of positive area (bottom) in pancreatic tumors from vehicle (*n* = 8) and Fasudil (*n* = 7)-treated mice. Box (upper and lower quartiles divided by median value) and whisker (5th–95th percentile) plots show outliers as individual points, with exact *P*-value by Mann–Whitney test.

D   Schematic diagram of the contribution made by ROCK signaling to pancreatic cancer invasive growth.

revealed that *Mmp10* and *Mmp13* were the most differentially expressed genes upon ROCK activation. Although MMP10 was not increased in cell lysates by ROCK activation (Fig 5B and D), both MMP10 and MMP13 were elevated in cell-conditioned medium (Fig 5B). Furthermore, when microvesicles were enriched by ultracentrifugation from conditioned medium, ROCK activation increased total protein (Fig 5C) and MMP10 content in this fraction (Fig 5D), with significantly greater ratios of MMP10 to total protein or MMP10 to caveolin, a lipid raft scaffolding protein found in microvesicles (Lee *et al*, 2011). Consistent with

our observations, MMP10 release has been reported to be mediated via microvesicles (de Lizarrondo *et al*, 2012) and the Rho-ROCK pathway was previously shown to contribute to microvesicle initiation by inducing membrane protrusions akin to blebs (Antonyak *et al*, 2012; Li *et al*, 2012). These results indicate that ROCK activation contributes to stromal remodeling through the combined effects of inducing changes in the expression of genes associated with ECM remodeling and increasing the efficient delivery of the corresponding encoded proteins to the proximal microenvironment.

ROCK activation also led to significant collagen degradation and remodeling associated with increased invasive growth (Figs 2, 3, 6 and 7). Importantly, treatment of ROCK-activated invasive cells with the MMP inhibitor GM6001 blocked invasion and reduced proliferation in three-dimensional collagen (Fig 7A–E), but not in two-dimensional contexts (Fig 7G and H), indicating that ROCK-induced collagen remodeling enables PDAC proliferation by relieving stromally imposed constraints (Fig 8D). Consistent with these observations, conditional ROCK activation *in vivo* reduced pancreatic cancer mouse survival (Fig 1H), while treatment with the ROCK inhibitor Fasudil prolonged survival and increased collagen levels (Fig 8A and C).

Despite much effort, new treatment regimens for pancreatic cancer have only minimally improved patient survival. The standard chemotherapeutic agent for PDAC is gemcitabine, either as a single agent or combined with albumin-bound paclitaxel, or a combination of fluorouracil, irinotecan, oxaliplatin, and leucovorin (FOLFIR-INOX) (Cid-Arregui & Juarez, 2015), which benefits a minority of patients and often leads to drug resistance over time. Our results indicate that ROCK inhibitors (Rath & Olson, 2012) as pancreatic cancer therapy might be beneficial due to reduced invasive growth of tumor cells due to the effect on impairing stromal collagen remodeling. In addition, these findings are in agreement with the concept that by reducing collagen remodeling activity it is possible to tip the balance such that dense collagen hinders, rather than promotes, pancreatic cancer. As with any systemically administered therapy, cell types in addition to tumor cells are potentially affected. Given that ROCK inhibition has previously been shown to block the activation of pancreatic stellate cells (Masamune et al, 2003) that have a tumor-promoting role in PDAC growth and progression (Apte et al, 2013), the therapeutic effect of Fasudil on PDAC mouse survival may be due to combined actions on tumor and non-tumor cells. We previously showed that conditional ROCK activation in mouse skin led to increased collagen deposition and accelerated progression of epidermal papillomas to invasive carcinomas (Samuel et al, 2011). The difference between these observations illustrates the context-dependence of the relationships between ROCK activation, collagen deposition/remodeling, and tumor progression.

## Materials and Methods

### Animal models

All mouse experiments were performed according to UK Home Office regulations. *Pdx1-Cre*, *LSL-KRas*$^{G12D}$, *LSL-Trp53*$^{fl}$, *and LSL-Trp53*$^{R172H}$ mice have been described previously (Jackson et al, 2001; Jonkers et al, 2001; Hingorani et al, 2003; Liu et al, 2004; Olive et al, 2004). Conditional *ROCK2* knockout (ROCK2$^{tm1a(KOMP)Wtsi}$) embryonic stem cells were supplied by the trans-NIH Knock-Out Mouse Project (KOMP, clone EPD0492_1_C01). Mice were subsequently crossed to *Actb* (*ActFLPe*) mice for removal of the lacZ/neo cassette, to *Pdx1-Cre* mice to generate pancreas-specific ROCK2 knockout mice, and to *Rosa26-LSL-RFP* reporter mice.

*LSL-ROCK:ER* mice were generated by gene-targeting to the *Hprt* locus (Fig EV2A) using an approach we have previously described (Samuel et al, 2009). Briefly, the sequence encoding the cytokeratin 14 promoter in the constructs pHprt$^{K14-ROCK:ER}$ and pHprt$^{K14-KD:ER}$

was replaced with a sequence encoding the chimeric synthetic promoter CAG (Niwa *et al*, 1991) and a floxed transcription termination sequence. The constructs were then used to separately target HM1 ES cells by electroporation, and targeted cells were selected in HAT medium as previously described (Samuel *et al*, 2009).

KPC experimental males were treated with 100 μl Fasudil (20 mg/ml) or water vehicle by gavage from 10 weeks of age until endpoint. RKPC experimental males were treated with Tamoxifen citrate salt in 1% EtOH (100 mg/l) or 1% EtOH vehicle in the drinking water from 10 weeks of age till endpoint, or for a maximum period of 9 weeks. Animals were sacrificed as per institutional guidelines. Organs and tumors were fixed in 10% formalin at room temperature and processed using standard histological methods.

### Human pancreas tissue

A commercial tissue microarray with normal pancreatic tissue and 78 cases of pancreatic cancer, including TNM, clinical stage, and pathology grade, was purchased from US Biomax (PA961c). Clinical data associated with these samples can be found in Appendix Table S4.

Tissue from border areas of resected tumors was collected prospectively following informed patient consent. The West of Scotland Research Ethics Committee 4 approved the study.

### Small molecules

Y27632 (Tocris 1254), H1152 (Tocris 2414), Fasudil (Selleckchem S1573, LC Laboratories F-4660), Blebbistatin (Tocris 1760), GM6001 (Millipore 142880-36-2), Tamoxifen citrate salt (Sigma T9262), 4HT (4-Hydroxytamoxifen, Sigma H7904).

### Antibodies

ROCK1 (BD-611136), ROCK2 (BD-610623), ROCK1/2 (Millipore 07-1458), phospho-MLC2 (Cell Signaling 3674), MRCL3/MRLC2/MYL9 (Santa Cruz sc-28329), MMP13 (Abcam ab39012), MMP10 (Leica NCL-MMP10), GFP (Abcam ab6556), RFP (Rockland 600-401-379), Ki67 (Vector VP-K452), Col1 3/4C (ImmuGlobe 0207-050), GAPDH (Millipore MAB374), Caveolin-1 (Santa Cruz sc-894), CD3 (Vector VP-RM01), CD31 (Abcam ab28364), α-Smooth Muscle Actin (Sigma-Aldrich A2547).

### Pancreatic tumor cell lines

Pancreatic ductal adenocarcinoma tumor cell lines were established from *Pdx1-Cre; LSL-KRas*$^{G12D/+}$; *LSL-Trp53*$^{R172H/+}$ (KPC) and *Pdx1-Cre; LSL-KRas*$^{G12D/+}$; *LSL-Trp53*$^{fl/+}$ (KPflC) mice (Morton et al, 2010). Retroviral pBABE puro constructs with conditionally active human ROCK1 (EGFP-ROCK1:ER), human ROCK2 (EGFP-ROCK2:ER), or GFP control (EGFP:ER) have been described previously (Croft et al, 2004). Stable KPflC cell lines were selected by standard procedures using 2.5 μg/ml Puromycin. PDAC cell lines were cultured in DMEM supplemented with 10% FBS, 2 mmol/l L-glutamine, and penicillin–streptomycin (complete DMEM). For experiments, cells were plated in complete DMEM or serum-free DMEM containing EtOH vehicle or 1 μM 4HT to activate ROCK kinase activity.

## Cell growth assay

The CellTiter-Glo Luminescent Cell Viability Assay (Promega) was used to determine cell growth in monolayer. $5 \times 10^3$ cells were plated into the wells of a 96-well cell culture dish. Wells were pre-coated with 0.05 mg/ml rat tail collagen type I (BD Biosciences) as indicated. Cells were left to settle and grow overnight in DMEM complete, and a first baseline measurement was taken. Then, 1 μM 4HT or EtOH vehicle was added to the medium, and second (24 h) and third (48 h) measurements were taken. The increase in viable cells at 24 h and 48 h time points was calculated relative to the baseline measurement.

## Immunoblotting

Standard protocols were used for Western blot analysis. Primary antibodies were routinely used at 1:500 or 1:1,000 dilutions. Alexa Fluor 680 and DyLight 800 (Thermo Fisher Scientific)-conjugated secondary antibodies were detected by infrared imaging (Li-Cor Odyssey). Pancreatic tissue was homogenized using the hard tissue homogenizing CK28-R Precellys lysing kit. Cells were grown in 6-well plates. Whole-cell lysates were prepared in cell lysis buffer (1% SDS, 50 mM Tris pH 7.5), and protein concentration was determined by Bicinchoninic assay (Sigma).

For validation of the conditional ROCK:ER activation in KPflC cell lines, $1 \times 10^6$ cells were plated in DMEM complete and allowed to settle and grow overnight. Next day, cells were washed 3× with serum-free DMEM and 2 ml serum-free DMEM with EtOH vehicle or 1 μM 4HT in the presence or absence of 1 μM or 10 μM H1152 was added to the cells. After 24 h of treatment, cell lysates were prepared for Western blot analysis.

For analysis of matrix metalloproteinase expression and release, $1 \times 10^6$ KPflC cells were plated in DMEM complete and allowed to settle and grow overnight. Next day, cells were washed 3× with serum-free DMEM and 2 ml serum-free DMEM with EtOH vehicle or 1 μM 4HT was added to the cells. After 48 h of treatment, cell lysates were prepared for Western blot analysis. Conditioned media were harvested at the same time. To remove cell debris, media were centrifuged 30 min at 10,000 × g, 4°C and the supernatant was snap frozen. For detection of MMP10, concentration of media was not required. For detection of MMP13, media were concentrated using 10K spin columns (Amicon Ultra-0.5 ml, UFC501096).

## Immunofluorescence

IF images were taken on a Zeiss 710 confocal microscope. Primary antibodies were routinely used at 1:50 or 1:100 dilutions. Alexa Fluor 488 phalloidin, Alexa Fluor 647 phalloidin, Alexa Fluor 488, and Alexa Fluor 594 (Thermo Fisher Scientific) secondary antibodies were used. Fluorescently labeled cells and tissue samples were mounted with ProLong Gold/ProLong Diamond incl. DAPI (Molecular Probes).

Cells were fixed in 4% paraformaldehyde (PFA) for 15 min and permeabilized with 0.5% Triton X-100/PBS for 5 min. Non-specific antibody binding was blocked by incubation of cells with 1% BSA/PBS for 5 min. Cells were then incubated with primary antibody for 1 h, followed by incubation with secondary antibody and/or phalloidin for 1 h.

To stain cells in three-dimensional collagen matrix invasion assays, PFA-fixed, paraffin-embedded sections were rehydrated and immersed in 10 mM citric acid buffer at pH 6.0, boiled for 20 min, cooled, and blocked with 10% normal goat serum in PBS. Sections were then incubated with primary antibody for 1 h, followed by incubation with Alexa Fluor 594 antibody for 1 h.

To stain cleaved collagen1 in three-dimensional collagen matrix invasion assays, PFA-fixed, paraffin-embedded sections were rehydrated and equilibrated in PBS. Sections were then incubated with collagen1 3/4C antibody for 1.5 h, followed by incubation with Alexa Fluor 594 antibody for 1 h.

## Immunohistochemistry

IHC slides were imaged and analyzed using the Leica SCN 400f scanner and Leica Slidepath Digital Image Hub software. Primary antibodies were routinely used at 1:50 or 1:100 dilutions. Formalin-fixed, paraffin-embedded sections were rehydrated and immersed in 10 mM citric acid buffer at pH 6.0, boiled for 20 min, cooled, and sequentially blocked with 3% $H_2O_2$ and 10% normal goat serum in PBS. Sections were then incubated with primary antibody, followed by incubation with Envision + System-HRP labeled Polymer (Dako). Staining was visualized with Liquid DAB + Substrate (Dako).

For collagen staining, formalin-fixed, paraffin-embedded sections were rehydrated and immersed in Picrosirius Red for 2 h.

To stain *Mmp13* in three-dimensional collagen matrix invasion assays, the RNAscope Probe Mm-Mmp13 (ACD 427601) was used according to manufacturer's instructions. Images were taken on an Olympus BX51 microscope.

To determine the ROCK2 and MMP10 histoscores of human pancreas adenocarcinoma cases, DAB staining intensities of pancreatic acinar cells (normal) or pancreatic tumor cells were scored. A weighted histoscore was calculated from the sum of ($1 \times$ % area of weak staining) + ($2 \times$ % area of moderate staining) + ($3 \times$ % area of strong staining), providing a semi-quantitative classification of staining intensity on a scale from 0 (negative) to 300 (strongest). Two researchers (N.R. and J.P.M.) quantified the TMA in a blinded fashion, and means were used for statistical analyses. Images of tissue microarrays were taken on an Olympus BX51 microscope.

## Microvesicle enrichment

Cells were cultured in serum-free DMEM containing either 1 μM 4HT or EtOH vehicle for 24 h. For each condition, 135 ml medium was collected and subjected to differential centrifugation and ultracentrifugation steps all performed at 4°C. Media were centrifuged at 300 × g for 10 min, followed by centrifugation at 2,000 × g for 10 min to pellet live and dead cells, respectively. Then, media were centrifuged in an ultracentrifuge (Optima L-90K, Beckman) at 10,000 × g for 30 min to remove cell debris, followed by ultracentrifugation at 100,000 × g for 70 min. The microvesicle pellet was washed with 36 ml PBS, followed by a second ultracentrifugation step at 100,000 × g for 70 min to re-pellet the microvesicles. The supernatant was discarded, and microvesicles were resuspended in 200 μl PBS. For Western blot analysis and Coomassie staining, 6× lysis buffer (1% SDS, 50 mM Tris pH 7.5) was added for protein extraction.

## Collagen degradation assay

Glass coverslips were placed on top of 25 μl Collagen type I, FITC conjugate 1 mg/ml (Sigma), and incubated at room temperature in the dark for 15 min. They were then placed on top of 50 μl 0.5% glutaraldehyde for 30 min. After three washes with PBS, $6 \times 10^4$ cells were plated on collagen-coated coverslips in 500 μl DMEM complete containing either 1 μM 4HT or EtOH vehicle in the absence or presence of the following drugs: 10 μM H1152 dihydrochloride, 10 μM GM6001, 50 μM Blebbistatin, or DMSO vehicle. Collagen and cells were fixed and IF stained after 16 or 18 h. Images of DAPI-stained cells and FITC-labeled collagen were taken on the Zeiss 710 confocal microscope. ImageJ was used to quantify the area of collagen degradation.

## Collagen matrix invasion assay

Collagen matrix invasion assays were performed as previously described (Timpson et al, 2011). Primary human fibroblasts were mixed with rat tail collagen1 and left in a cell culture incubator for a week to allow conditioning of the collagen. To remove fibroblasts from the collagen matrices, the disks were incubated with 5 μg/ml Puromycin for at least 24 h and then washed twice with medium. $2 \times 10^5$ cells were seeded in DMEM complete without or with 1 μM 4HT on top of the disks and allowed to settle and grow over 2 days. Afterward, the collagen matrices were mounted onto grids to generate an air/liquid interface. DMEM complete containing 1 μM 4HT, 10 μM H1152, 10 μM GM6001, or DMSO was added to the dishes as indicated. After 8 days of invasion, the collagen matrix disks were fixed in 4% paraformaldehyde overnight and processed using standard histological methods.

H&E-stained sections were scanned and analyzed using Digital Slide Server (Slidepath) software. For quantification of invading cells, the polygon tool was used to draw a rectangle covering the whole collagen matrix disk but omitting the stationary cells at the surface. An algorithm was written to specifically detect H&E-stained PDAC cells in collagen matrices. The area of invading cells was divided by the total area and percentages were used to calculate the invasion index, relative to the respective control, for graphs and statistics.

Immunofluorescence-stained sections were analyzed with the ImageJ cell counter plugin. 4–10 immunofluorescence images were taken at 40× magnification of each disk. For graphs and statistics, total counts of cells at the surface or percentages of Ki67-positive cells over total cell number were used.

## Determination of collagen quantity and quality

Collagen second harmonic images were collected using a LaVision Biotec Trim-scope equipped with a Coherent Chameleon Ti: Sapphire femtosecond pulsed laser and analyzed in ImageJ as described in (Miller et al, 2015). An excitation wavelength of 890 nm was used so that the second harmonic generation (SHG) would be generated at a central wavelength of 445 nm and focused to the sample plane by a long working distance 20× (NA = 0.95) water immersion objective from Olympus. A z-stack of 100 μm deep was imaged over a region of 500 μm by 500 μm, with at least three duplicates of each condition. The UMB GLCM plugin was used as the basis for the texture analysis, but modified so as to run automatically through the four directions of comparison, for each of the 100 comparison distances. Firstly, the user selected a directory containing the collagen stack images. A maximum projection image was then produced and duplicated. The duplicate image was then automatically thresholded to produce a mask that was then applied to the original maximum projection image. This removed the background noise bias introduced in the GLCM analysis by selecting only the collagen SHG signal. The masked image was then passed to the GLCM texture plugin (Kvaal et al, 2008), which had been modified so that it could be operated in a nested loop for varying pixel comparison distances and alternative directions of comparison. The output of the plugin for each image was 100 rows of the five texture parameters over each of four directions, totaling 2,000 parameter values. These were saved as a text data file for each image. When all the images in the directory were analyzed, the data files were processed using a MATLAB script that produced the mean of each texture parameter for each image. These were then imported into Prism, where a double exponential decay model was fit to the data for the correlation GLCM parameter and the weighted mean decay distance for each sample was calculated in Excel.

## Transmission electron microscopy

For contrast staining, ultrathin collagen matrix sections were incubated with 2% methanolic uranyl acetate for 5 min and then Reynolds' lead citrate for 5 min. Collagen was visualized by incubation with 2% aqueous uranyl acetate for 10 min, 2% aqueous tannic acid for 10 min, and then Reynolds' lead citrate for 5 min.

Ultracentrifuge-pelleted microvesicles were fixed in 2% PFA/PBS for 1 h at 4°C. All other processing steps were performed at room temperature. 5 μl droplets of microvesicles were placed onto 300mesh Formvar/carbon-coated Nickel grids and left to settle for 30 min. For routine negative staining, grids were rinsed by floating sample side down on droplets of distilled water ($6 \times 2$ min) before staining with 2% Ammonium Molybdate for 30 s, and then left to dry before imaging. For immunogold labeling, grids were floated sample side down on droplets of PBS ($6 \times 5$ min), followed by droplets of 0.05 M Glycine/PBS ($4 \times 5$ min) and droplets of 3% BSA/PBS ($6 \times 5$ min). Next, grids were transferred onto droplets of MMP10 primary antibody in 3% BSA/PBS (1:50 dilution) for 1 h. Grids were passed through droplets of 3% BSA/PBS ($6 \times 5$ min) and then incubated on droplets of gold conjugate GAM (1:20 dilution) for 1 h. Grids were washed with 3% BSA/PBS ($6 \times 5$ min), PBS ($6 \times 5$ min), fixed with 1% Glutaraldehyde/PBS for 5 min, and rinsed 10× in droplets of distilled water. Grids were transferred onto droplets of Uranyl-Oxalate solution (pH 7) for 5 min and then onto droplets of Methyl Cellulose-UA for 10 min on ice. Finally, grids were scooped up on platinum loops and left to dry after excess Methyl Cellulose-UA was carefully removed with filter paper (Théry et al, 2001).

Images were viewed on the FEI Tecnai T20 transmission electron microscope running at 200 kV, and images were captured using a GATAN Multiscan camera 794 and GATAN Digital Imaging software.

    

**The paper explained**

**Problem**

Pancreatic ductal adenocarcinoma (PDAC) is the most common type of pancreatic cancer, which is one of the leading causes of cancer death. Despite some recent advances in treatment regimens, the overall outcome is still poor with a five-year survival of < 5%. Consequently, there is a significant unmet clinical need for the identification of alternative therapeutic targets.

**Results**

We show that ROCK kinase expression is up-regulated during pancreatic cancer progression and that high ROCK levels correlate with decreased survival time. Cell-based assays demonstrate that ROCK activation is sufficient to drive PDAC cell invasion and growth in 3D collagen matrices. We identified ROCK signaling as a driver of a gene expression program leading to extracellular matrix remodeling and collagen degradation through expression and release of matrix metalloproteinases via microvesicles. Importantly, ROCK inhibitors reversed these effects and significantly prolonged PDAC mouse survival.

**Impact**

Our results highlight an ancillary role for elevated ROCK signaling in pancreatic tumor growth and progression. We suggest that ROCK kinases should be considered as a potential target for combination chemotherapy of invasive pancreatic cancer.

**RNA sequencing and quantitative polymerase chain reaction**

$1 \times 10^6$ KPflC cells were seeded into 6-well plates in DMEM complete and allowed to settle and grow overnight. Next day, cells were washed three times with serum-free DMEM and 2 ml serum-free DMEM with drugs was added: 1 μM 4HT or 1 μM H1152. After 24 h of treatment, total RNA was extracted from cells with the RNAeasy kit (Qiagen) and RNA was quantified using a nanodrop spectrophotometer (NanoDrop Tech).

For RNA sequencing, RNA quality was assessed using the Agilent 2100 Bioanalyzer with the RNA 6000 Nano LabChipVR reagent set (Agilent Technologies). The Illumina TruSeq RNA Library Preparation kit v2.0 was used to prepare an oligo dT-based library. The library was sequenced on the NextSeq 500 platform using the High Output 75 cycles kit (2 × 36 cycles, paired-end reads, single index). Quality control of raw RNASeq data files was performed by fastqc (http://www.bioinformatics.babraham.ac.uk/projects/fastqc/) and fastq_screen (http://www.bioinformatics.babraham.ac.uk/projects/fastq_screen/). Then, RNASeq reads were aligned to the mouse genome (GRCm38.75) using TopHat2 (Kim *et al*, 2013) and resulting bam files processed with htseq_count (http://www.huber.embl.de/users/anders/HTSeq/doc/count.html). The final counts were normalized and analyzed with DESeq2 (Love *et al*, 2014). Statistically significant differences in gene expression were determined with a false discovery rate (FDR) of 5%.

For quantitative polymerase chain reaction (qPCR), complementary DNA was synthesized using the QuantiTect Reverse Transcription Kit (Qiagen). Quantitative reverse transcriptase (RT)–PCR primers for Mmp10, Mmp13 and Gapdh were acquired from Qiagen (Quantitect Primer Assay). QPCR was set up with the DyNAmo HS SYBR Green qPCR Kits (Thermo Fisher Scientific) and run on the 7500 Fast Real-Time PCR System (Applied Biosystems).

**Statistical analyses**

Calculations were done in GraphPad Prism, and the statistical tests used are indicated in each figure legend. Bar graphs represent means ± SEM. Box (upper and lower quartiles divided by median value) and whisker (5th–95th percentile) plots show outliers as individual points.

**Expanded View** for this article is available online.

**Acknowledgements**

Funding from Cancer Research UK (A18276). N.R. received a Cancer Research UK Research Travel Award and a Company of Biologists Travelling Fellowship. Thanks to William Clark for RNA sequencing, and the Beatson Institute Histology and Biological services.

**Author contributions**

NR performed experiments and analyzed data; JPM, LJ, LH, SK, EJM, MM, and AVP contributed to experiments; MSS established the *LSL-ROCK:ER* mouse line; GK analyzed RNA seq data; IR advised on primary acinar cell culture experiments; KIA advised on second harmonic generation microscopy and analysis; MFO supervised the work and assisted with data analysis; NR and MFO prepared the figures and wrote the manuscript.

**Conflict of interest**

The authors declare that they have no conflict of interest.

**For more information**

Unprocessed RNA sequencing reads have been deposited as fastq files at the National Center for Biotechnology Information (NCBI) Sequence Reads Archive (SRA) with the reference SRP081135 (https://www.ncbi.nlm.nih.gov/sra/?term=srp081135) (Rath *et al*, 2016). In addition, a project overview has been submitted as the BioProject reference PRJNA327913 (http://www.ncbi.nlm.nih.gov/bioproject/?term=PRJNA327913) with a description of the BioSample reference SAMN05361890 (http://www.ncbi.nlm.nih.gov/biosample/?term=SAMN05361890).

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
