## [Review Process File · EMBO Molecular Medicine]

ROCK signaling promotes collagen remodeling to facilitate invasive pancreatic ductal adenocarcinoma tumor cell growth

Nicola Rath, Jennifer P Morton, Linda Julian, Lena Helbig, Shereen Kadir, Ewan J McGhee, Kurt I Anderson, Gabriela Kalna, Margaret Mullin, Andreia V Pinho, Ilse Rooman, Michael S Samuel, and Michael F Olson

Corresponding author: Michael Olson, Cancer Research UK Beatson Institute

Review timeline:

Submission date:	23 June 2016
Editorial Decision:	02 August 2016
Revision received:	21 October 2016
Editorial Decision:	21 November 2016
Revision received:	24 November 2016
Accepted:	28 November 2016

Transaction Report:

Editor: Roberto Buccione

1st Editorial Decision

02 August 2016

Thank you for the submission of your manuscript to EMBO Molecular Medicine. We are very sorry that it has taken so long to get back to you on your manuscript.

In this case we experienced unusual difficulties in securing three willing and appropriate reviewers. Further to this, Reviewer #2 ultimately failed to deliver his/her report. As a further delay cannot be justified I have decided to proceed based on the two available consistent evaluations.

As you will see, both Reviewers are largely positive although they raise a number of important and partially overlapping issues that require your action. I will not go into much detail, as their comments are quite clear.

Reviewer 1 would like to see more information on the cancer samples and also wonders whether the ROCK inhibitor would inhibit the ROCK-driven PDAC cell invasion and proliferation. The reviewer would also like you to investigate the presence of MMP13 in MVs and at the invasive interfaces. Reviewer 1 is also asking for substantial clarification on a number of experimental approaches.

Reviewer 3, similarly to Reviewer 1, would like to know more about the clinical history on the samples in the TMA and notes the lack of ROCK loss-of-function experiments. S/he is also not convinced that the current data demonstrate MMP10 in the MVs. Another important point is the lack

of cross-validation of the mouse findings in the TMA. Reviewer 3 also lists a few other issues that require attention.

While publication of the paper cannot be considered at this stage, we would be pleased to consider a revised submission, with the understanding that the Reviewers' concerns must be fully addressed including with additional experimental data where appropriate and that acceptance of the manuscript will entail a second round of review.

Please note that it is EMBO Molecular Medicine policy to allow a single round of revision only and that, therefore, acceptance or rejection of the manuscript will depend on the completeness of your responses included in the next, final version of the manuscript.

As you know, EMBO Molecular Medicine has a "scooping protection" policy, whereby similar findings that are published by others during review or revision are not a criterion for rejection. However, I do ask you to get in touch with us after three months if you have not completed your revision, to update us on the status. Please also contact us as soon as possible if similar work is published elsewhere.

Please note that EMBO Molecular Medicine now requires a complete author checklist (<http://embomolmed.embopress.org/authorguide#editorial3>) to be submitted with all revised manuscripts. Provision of the author checklist is mandatory at revision stage; The checklist is designed to enhance and standardize reporting of key information in research papers and to support reanalysis and repetition of experiments by the community. The list covers key information for figure panels and captions and focuses on statistics, the reporting of reagents, animal models and human subject-derived data, as well as guidance to optimise data accessibility. In this case, the author checklist is especially relevant as, in addition to the concerns on the clinical features of the TMA, I note that both reviewers have reservations on your presentation of statistics information. The Author checklist will be published alongside the paper, in case of acceptance, within the transparent review process file

Please note that we now mandate that all corresponding authors list an ORCID digital identifier. You may do so through our web platform upon submission and the procedure takes <90 seconds to complete. We also encourage co-authors to supply an ORCID identifier, which will be linked to their name for unambiguous name identification.

I look forward to seeing a revised form of your manuscript as soon as possible.

***** Reviewer's comments *****

Referee #1 (Comments on Novelty/Model System):

Appropriate model systems have been used.

Referee #1 (Remarks):

The manuscript entitled "ROCK signaling promotes collagen remodeling to facilitate invasive pancreatic ductal adenocarcinoma tumor cell growth" elucidated that the expression of ROCK1 and ROCK2 kinases was increased with tumor progression and reduced survival. Conditional ROCK1 or ROCK2 activation promoted collective invasion and proliferation in 3-dimensional collagen matrices which can be blocked by MMP inhibition. These findings revealed an ancillary role for increased ROCK signaling in advanced pancreatic cancer which enables invasive tumor growth by overcoming microenvironmentally-imposed proliferation restraints. An implication of these results is that ROCK inhibitor administration to pancreatic cancer patients might reverse the ability of pancreatic cancer cells to surmount the growth-restraining properties of tumor-associated desmoplasia. The study was well designed and the conclusion is well supported by the data.

Minor concerns:

1. Fig. 1E, the ROCK1 and ROCK2 show strongly coordinated expression pattern in TCGA. Why is that?

2. Usually 4-HT treatment should only activate the fusion protein ROCK1/2:ER, not induce the expression level of ROCK1/2:ER, but it seems there is significant increase of ROCK1/2:ER upon 4HT treatment in the blots in Fig. 2B, esp. ROCK2:ER infected KPflC cells. Please clarify.
3. What is the difference of experimental conditions for Fig. 2C vs 2D? Why there is significantly higher invading cell area in Fig. 2C than 2D, for both Rock1:ER (6% vs 3%) and ROCK2:ER(4% vs 2%) expressing cells?
4. It would strengthen the 4HT activation of ROCK1/2:ER protein by showing the nuclei distribution in FL images of KPflC cells expressing GFP-ER, ROCK1:ER or ROCK2:ER fusion protein, since all these were fused with GFP already.
5. In Fig. 4B and 4D, both MMP10 and MMP13 show significant increase in conditional medium in 4HT treated KPflC cell expressing ROCK1/2:ER, and more MMP10 was found in the microvesicle fraction, what about MMP13, is that also increased in microvesicle?
6. What is the genotype of the Normal tissue in Fig 1F, is it from KC or KPC mouse, and the age of mice for this normal tissue?
7. There is one typo in Fig. 1 figure legend: the (I) should be (H).
8. Keywords list should be added.
9. The median survival period and exact p values should be described in the text accordingly.
10. Figure 8, the word "tumor" should be consistently used. In addition, the figure should be modified to be more informative and specific.
11. ROCK2 expression was found to increase with pancreatic cancer progression both in human and KrasG12D-driven mouse tumors. The author needs to show the clinical information of 78 pancreas adenocarcinoma cases. Did the author statistically analyze the correlation between expression of ROCK2 and clinical characteristics such as tumor stage?
12. In Fig. 2 the author studied the association between ROCK-induced invasion and collagenolysis and found collagen 1 cleavage was minimal in GFP:ER expressing cells, comparing with invasive ROCK1:ER and ROCK2:ER cells. Did the author detect other type of collagens?
13. In Fig.2 ROCK kinases drive PDAC cell invasion and proliferation when treated with 4-HT. Can the phenotypes be reversed by the ROCK selective inhibitor H1152?
14. In Fig.6D immunofluorescence revealed that ROCK-induced MMP10 was present at invasive cell interfaces with collagen matrix. What about MMP13?

Referee #3 (Remarks):

This paper presents interesting data around the model that ROCK kinases drive MMP3 and MMP10 expression, which are secreted in vesicles and lead to collagen lysis and increased invasion. The authors begin with an analysis of expression and survival data in human studies and show recapitulation in the KPC mouse. The subsequent experiments center on cell lines with inducible ROCK1 and ROCK2 kinase domains, demonstrating a variety of effects in support of their model. Overall the experiments suggest importance of this pathway and are supportive of the conclusions. The strong staining in blebs of MMP10 is particularly noteworthy. But I believe a few important points should be addressed to strengthen the paper.

The IHC data in fig. 1A is not convincing - total staining in the images looks similar. Only scant information is given about the histoscore. An unbiased quantification, or blinded analysis by 2 pathologists, would be better.

Fig. 1A - did the stage information come from Biomax, who supplied the TMA? It would be good to know who determined stage and based on what criteria (e.g. AJCC guidelines?). The anatomical stage is not necessarily associated with the grade of differentiation of the primary tumor; for example some stage IV cancers have well-differentiated ducts. Thus the authors should determine whether ROCK expression in the primary tumors is related to histomorphology or histological grade.

Further, it seems the findings from the mouse studies should be tested in the human sections. Does the IHC in the TMA show evidence of increased staining at the invasive front, or in blebs?

In the treatment of the kpc mice with fasudil, the authors show evidence of reduced collagen without the ROCK inhibitor (fig. 7a). If the proposed mechanism is increased invasion through collagen lysis (fig. 7c), then they also should see reduced metastases with fasudil, or other evidence of

reduced invasion. It seems like further characterization of the mice would help understand the contribution of ROCK to pancreatic cancer.

The conclusion from fig. 5 that MMP10 is carried in microvesicles is not entirely clear. Since the level in total media is higher (fig. 7b), any residual media would cause elevated levels. Also, the authors haven't proven that they have isolated vesicles. Fig. 5c shows higher protein content, but are these vesicle proteins? Perhaps a comparison should be made to the non-vesicle fraction for total protein but also vesicle-specific proteins. Since they have access to EM, immune-gold visualization of vesicles would be a direct way to look at MMP10 in vesicles.

The authors base nearly all experiments on upregulation of ROCK. Key experiments to test their hypothesis would be knockdown/knockout of ROCK in invasive cells that express it.

Additional comments:

Fig. 1F - do the authors have data on ROCK activation in the KC mice? That would be a good comparison also.

Fig. 2B - It does not appear that the ROCK1/2:ER bands decrease upon treatment with H1152, as the authors claim.

Minor:

The p values in fig. 1 have 3 significant digits, which likely is too much precision for the number of samples given.

1st Revision - authors' response

21 October 2016

Thanks to both reviewers for their positive comments and helpful suggestions. Our point by point responses are provided below.

Referee #1 (Comments on Novelty/Model System):

Appropriate model systems have been used.

Referee #1 (Remarks):

The manuscript entitled "ROCK signaling promotes collagen remodeling to facilitate invasive pancreatic ductal adenocarcinoma tumor cell growth" elucidated that the expression of ROCK1 and ROCK2 kinases was increased with tumor progression and reduced survival. Conditional ROCK1 or ROCK2 activation promoted collective invasion and proliferation in 3-dimensional collagen matrices which can be blocked by MMP inhibition. These findings revealed an ancillary role for increased ROCK signaling in advanced pancreatic cancer which enables invasive tumor growth by overcoming microenvironmentally-imposed proliferation restraints. An implication of these results is that ROCK inhibitor administration to pancreatic cancer patients might reverse the ability of pancreatic cancer cells to surmount the growth-restraining properties of tumor-associated desmoplasia. The study was well designed and the conclusion is well supported by the data.

Minor concerns:

1. Fig. 1E, the ROCK1 and ROCK2 show strongly coordinated expression pattern in TCGA. Why is that?

RESPONSE: This is an interesting question. Our hypothesis is that conditions that respond in a positive way to ROCK signaling exert a selective advantage to cells/tumors expressing both ROCKs in a coordinated manner. This may have to do with the way that both genes are transcriptionally regulated. Consistent with this hypothesis, increased ROCK signaling was shown to make a positive contribution to pancreatic cancer recently in Laklai et al. (2016 Nature Med 22 p497-505). A brief discussion of this has been added to the final sentence at the bottom of manuscript page 6.

2. Usually 4-HT treatment should only activate the fusion protein ROCK1/2:ER, not induce the expression level of ROCK1/2:ER, but it seems there is significant increase of ROCK1/2:ER upon 4HT treatment in the blots in Fig. 2B, esp. ROCK2:ER infected KPflC cells. Please clarify.

RESPONSE: The reviewer is correct that 4-HT increases the amount of ROCK:ER proteins as shown in the re-numbered Figure 2C, an observation we've consistently seen and have reported previously (e.g. Croft et al. 2004 *Cancer Res* 64 p8994-9001). Although in some instances, 4-HT activation of ER fusion proteins works by directly increasing catalytic activity, for example Raf1:ER (Samuels & McMahon. 1994 *Mol Cell Biol* 14 p7855-7866), in other cases activation apparently works entirely through 4HT-induced protein accumulation, for example MEK1:ER (Greulich & Erikson. 1998 *J Biol Chem* 273 p13280-13288). For ROCK1:ER and ROCK2:ER, it appears as though 4-HT-induced conditional activation works via a combination of increased specific activity as well as protein accumulation, although this has not been examined in great deal. The observation that 4HT treatment resulted in higher levels of ROCK:ER fusion proteins has now been mentioned in the manuscript at the top of manuscript page 9.

3. What is the difference of experimental conditions for Fig. 2C vs 2D? Why there is significantly higher invading cell area in Fig. 2C than 2D, for both Rock1:ER (6% vs 3%) and ROCK2:ER(4% vs 2%) expressing cells?

RESPONSE: Primary human fibroblasts were used in each experiment to condition the collagen matrix prior to seeding the PDAC tumor cells on top, as shown in the re-numbered Figure 2D (formerly Figure 2C). Depending on the source, batch and passage number of the fibroblasts, the collagen density of the matrices after conditioning may vary due to differences in re-organization and contraction of the collagen bundles. This will have an influence on the extent of tumor cell invasion. However, we consistently see comparable relative levels of invasion between experiments. To take into account variation in the properties of the fibroblast-conditioned collagen matrices between experiments separated by time, we made our comparisons between conditions (e.g. 4HT treated GFP:ER vs ROCK1:ER or ROCK2:ER) within the same sets of experiments. In addition, to make the results of these experiments clearer and easier to compare, we have converted the absolute values of "invading cell area" to a normalized "invasion index" as described in the Methods section.

Please note that the previous Fig. 2D has been replaced by new data in Figure EV3B.

4. It would strengthen the 4HT activation of ROCK1/2:ER protein by showing the nuclei distribution in FL images of KPfIC cells expressing GFP-ER, ROCK1:ER or ROCK2:ER fusion protein, since all these were fused with GFP already.

RESPONSE: We agree with this suggestion that it would be useful to visualize the 4HT-induced activation of ROCK1/2:ER using the GFP fluorescent channel, but unfortunately the fusion of ROCK1:ER or ROCK2:ER to GFP reduces the fluorescence signal intensity to undetectable levels. We've not been able to detect GFP fluorescence of these fusion proteins in any of our previously published studies, so it's not unique to the systems used in this manuscript.

5. In Fig. 4B and 4D, both MMP10 and MMP13 show significant increase in conditional medium in 4HT treated KPfIC cell expressing ROCK1/2:ER, and more MMP10 was found in the microvesicle fraction, what about MMP13, is that also increased in microvesicle?

RESPONSE: As the reviewer is undoubtedly aware, not all antibodies perform equally well in different applications. The monoclonal MMP10 antibody is excellent; we became aware of it from the publication by Briso et al. (2013 *Genes & Dev* 27 p1959-1973). Unfortunately, we have tested 3 different MMP13 antibodies and none performs as well or is as sensitive as the MMP10 antibody. We have now added immunofluorescence images revealing the increased abundance of MMP13 in bleb-like protrusions of ROCK1:ER and ROCK2:ER cells with 4HT treatment (Figure 5A, right panels and Figure EV4B), which parallels the observed increase and localization of MMP10 (Figure 5A, left panels and Figure EV4A). Although we previously were able to show the ROCK activity induced increase in MMP13 in conditioned medium in Figure 5B, the tiny amount of material isolated after microvesicle isolation did not allow us to detect an MMP13 signal with the antibodies tested.

6. What is the genotype of the Normal tissue in Fig 1F, is it from KC or KPC mouse, and the age of mice for this normal tissue?

RESPONSE: The normal pancreas tissues in Figures 1F and 1G were taken from wildtype mice, and the labelling of Figure 1F has been changed to clarify this point. The wild-type mice ranged in age from 30 to 357 days.

7. There is one typo in Fig. 1 figure legend: the (I) should be (H).

RESPONSE: Thanks for pointing out this typo, which has been corrected.

8. Keywords list should be added.

RESPONSE: A keyword list has been added.

9. The median survival period and exact p values should be described in the text accordingly.

RESPONSE: Both are now mentioned for our survival studies in the main text.

10. Figure 8, the word "tumor" should be consistently used. In addition, the figure should be modified to be more informative and specific.

RESPONSE: We have corrected the spelling to "tumor", and have adjusted the schematic diagram to be more informative.

11. ROCK2 expression was found to increase with pancreatic cancer progression both in human and KrasG12D-driven mouse tumors. The author needs to show the clinical information of 78 pancreas adenocarcinoma cases. Did the author statistically analyze the correlation between expression of ROCK2 and clinical characteristics such as tumor stage?

RESPONSE: We have added the clinical information data provided by the TMA supplier US Biomax (Supplemental Table S5) and we show the correlation between ROCK2 histoscore and tumor stage (Figure 1A), and ROCK2 histoscore and tumor grade (Figure EV1A).

12. In Fig. 2 the author studied the association between ROCK-induced invasion and collagenolysis and found collagen 1 cleavage was minimal in GFP:ER expressing cells, comparing with invasive ROCK1:ER and ROCK2:ER cells. Did the author detect other type of collagens?

RESPONSE: We also tested the ability of these cells to degrade gelatin but GFP:ER, ROCK1:ER or ROCK2:ER expressing cells did not show much of an effect, so we did not continue with these experiments.

13. In Fig.2 ROCK kinases drive PDAC cell invasion and proliferation when treated with 4-HT. Can the phenotypes be reversed by the ROCK selective inhibitor H1152?

RESPONSE: We have added new data confirming that H1152 ROCK inhibitor strongly inhibits cell invasion and proliferation in ROCK1:ER and ROCK2:ER expressing cells (Figure EV3B-E). In addition, the new Figure 2A shows that invasive KPC cells lose their ability to invade into collagen matrix in presence of H1152.

14. In Fig.6D immunofluorescence revealed that ROCK-induced MMP10 was present at invasive cell interfaces with collagen matrix. What about MMP13?

RESPONSE: As mentioned above, the MMP10 antibody worked very well in all applications, while finding good MMP13 antibodies was a consistent challenge. We could not identify an antibody that specifically detects MMP13 in cells that are embedded in collagen matrices, neither by immunofluorescence nor by immunohistochemistry. Nevertheless to answer your question, we have now added new data in Figure 6E that shows immunohistochemistry-stained *Mmp13* using an RNAscope *in situ* hybridization probe. The images reveal an upregulation of *Mmp13* RNA expression in ROCK1:ER and ROCK2:ER cells at the invading front and collagen interfaces.

Referee #3 (Remarks):

This paper presents interesting data around the model that ROCK kinases drive MMP3 and MMP10 expression, which are secreted in vesicles and lead to collagen lysis and increased invasion. The authors begin with an analysis of expression and survival data in human studies and show recapitulation in the KPC mouse. The subsequent experiments center on cell lines with inducible ROCK1 and ROCK2 kinase domains, demonstrating a variety of effects in support of their model. Overall the experiments suggest importance of this pathway and are supportive of the conclusions. The strong staining in blebs of MMP10 is particularly noteworthy. But I believe a few important points should be addressed to strengthen the paper.

The IHC data in fig. 1A is not convincing - total staining in the images looks similar. Only scant information is given about the histoscore. An unbiased quantification, or blinded analysis by 2 pathologists, would be better.

RESPONSE: The IHC images, which were originally taken with a Leica SCN 400f scanner, have now been replaced with images taken on an Olympus BX51 microscope to provide better contrast that is more representative. More details about the histoscore calculation have been added to the Methods section. ROCK2 expression was determined in pancreatic acinar cells for normal tissue or in tumor cells for adenocarcinoma tissue, while other cell populations (e.g. invading fibroblasts) were not included. Two experienced pancreatic cancer researchers quantified the TMA independently and blinded.

Fig. 1A - did the stage information come from Biomax, who supplied the TMA? It would be good to know who determined stage and based on what criteria (e.g. AJCC guidelines?). The anatomical stage is not necessarily associated with the grade of differentiation of the primary tumor; for example some stage IV cancers have well-differentiated ducts. Thus the authors should determine whether ROCK expression in the primary tumors is related to histomorphology or histological grade.

RESPONSE: The human pancreatic cancer tissue array was purchased from US Biomax, which was packaged together with the clinical information data that is now provided as Supplemental Table S5. In addition to tumor stage, we now show the correlation between ROCK2 histoscore and tumor grade as well (Figure EV1A).

Further, it seems the findings from the mouse studies should be tested in the human sections. Does the IHC in the TMA show evidence of increased staining at the invasive front, or in blebs?

RESPONSE: We stained a selection of human PDAC tissues focusing on the resection margin. As shown in new Figure EV1B, ROCK2 is expressed in cells in the resection margin although is not greatly increased at the invasive front. Unfortunately, the small size of blebs on individual cells does not allow enough spatially resolved detail to distinguish individual blebs in our IHC stained tissue images. To further correlate our findings in mouse studies to human pancreatic cancer, we stained the pancreatic cancer tissue array from US Biomax for MMP10 and found a positive correlation between higher MMP10 histoscore and higher tumor grade as shown in a new Figure 4G.

In the treatment of the kpc mice with fasudil, the authors show evidence of reduced collagen without the ROCK inhibitor (fig. 7a). If the proposed mechanism is increased invasion through collagen lysis (fig. 7c), then they also should see reduced metastases with fasudil, or other evidence of reduced invasion. It seems like further characterization of the mice would help understand the contribution of ROCK to pancreatic cancer.

RESPONSE: This experiment was designed to reveal the influence of ROCK inhibitor on survival rather than metastasis. Consequently, the metastatic burden was only analyzed at the clinical endpoint that was delayed by a median of 45 days in fasudil-treated KPC mice compared to vehicle-treated KPC mice. Keeping in mind that the more aged mice might be expected to have an increased incidence of metastasis, we found equal incidences of metastasis to the lung or liver in control and fasudil treated mice. Although consistent with the possibility that the fasudil effect on increasing collagen reduced the potential for metastasis to occur over the longer lifespan of these mice, the experiment wasn't formally designed as a study on metastasis. For further characterization, we have added a panel of analyses done on the KPC endpoint tumors in a new Appendix Figure S8.

The conclusion from fig. 5 that MMP10 is carried in microvesicles is not entirely clear. Since the level in total media is higher (fig. 7b), any residual media would cause elevated levels. Also, the authors haven't proven that they have isolated vesicles. Fig. 5c shows higher protein content, but are these vesicle proteins? Perhaps a comparison should be made to the non-vesicle fraction for total protein but also vesicle-specific proteins. Since they have access to EM, immune-gold visualization of vesicles would be a direct way to look at MMP10 in vesicles.

RESPONSE: The isolation of microvesicles was performed using established methods that incorporate washing with large excesses of wash buffer. As a result, we are confident that carry-over from medium would not have significantly contributed to the increased protein observed in Figure 5C. Nevertheless, we now provide new data in a revised Figure 5D with TEM pictures of microvesicles from the very same samples that were used for total microvesicle protein gels as well as MMP10 and Caveolin western blots. The first EM picture (routine negative staining) confirms that we have isolated microvesicles using the classical method of differential centrifugation, and the second EM picture (immunogold staining) validates that MMP10 is actually present in our isolated microvesicles. Furthermore, the western blot in Figure 5D shows the presence of caveolin in these

microvesicle fractions, a protein previously reported to be present in microvesicles (e.g Di Vizio et al. 2009 Cancer Res 69 p 5601-5609; Logozzi et al. 2009 PLoS One 4 p e5219).

The authors base nearly all experiments on upregulation of ROCK. Key experiments to test their hypothesis would be knockdown/knockout of ROCK in invasive cells that express it.

RESPONSE: We have added new results from four more experiments to confirm that inhibition of ROCK signaling has the reverse effect relative to activation of ROCK signaling. First, the invasion of KPC PDAC cells was significantly inhibited by H1152 ROCK inhibitor (Figure 2A). Second, ROCK1/2:ER-dependent invasion and proliferation of KPfC cells was significantly inhibited by H1152 ROCK inhibitor (Figure EV3B-E). Third, ROCK1/2:ER-dependent blebbing and accumulation of MMP10 and MMP13 in membrane protrusions was not apparent in H1152 treated cells (Figure EV4A and EV4B). Fourth, while activation of ROCK2:ER decreased the attachment of primary acinar cells, Y27632 ROCK inhibitor increased their attachment (Appendix Figure S2D and S2E).

Additional comments:

Fig. 1F - do the authors have data on ROCK activation in the KC mice? That would be a good comparison also.

RESPONSE: We tested the effect of ROCK2:ER activation during PanIN formation in KC mice (Appendix Figure S3). Because our analysis did not reveal any differences at these early stages, these studies were not continued further.

Fig. 2B - It does not appear that the ROCK1/2:ER bands decrease upon treatment with H1152, as the authors claim.

RESPONSE: To clarify, the point made in the description of the results in Figure 2C (formerly Figure 2B) was that H1152 decreased the induction by 4HT of increased MLC phosphorylation in ROCK1:ER and ROCK2:ER expressing cells. This sentence has been re-written to make this point clearer.

Minor:

The p values in fig. 1 have 3 significant digits, which likely is too much precision for the number of samples given.

RESPONSE: All reported p values have been adjusted to 2 significant digits.

2nd Editorial Decision

21 November 2016

Thank you for the submission of your revised manuscript to EMBO Molecular Medicine.

We have now received the enclosed report from the reviewer #3 who was asked to re-assess it. As you will see the reviewer is now globally supportive and I am pleased to inform you that we will be able to accept your manuscript pending the following final amendments. Please note that I had also asked this reviewer to evaluate your responses to reviewer #1 as well, as I could not reach him/her.

Please add the information that two researchers quantified the TMA in a blind fashion to the manuscript text as suggested by the reviewer.

Please submit your revised manuscript within two weeks. I look forward to seeing a revised form of your manuscript as soon as possible.

***** Reviewer's comments *****

Referee #3 (Remarks):

The authors have done a good job addressing all comments and providing new data, figures, and text to strengthen the paper. The staining data (Fig. 1) are certainly more convincing. The new experiments testing the effects of ROCK inhibition also are quite convincing. Thus I think the paper should be a useful contribution to the field. I have one minor comment that I suggest they address.

Remaining comment:

Regarding the analysis of the IHC: The authors state in their responses that two researchers quantified the TMA and were blinded. The authors should state these points in the methods section, and I think it would be appropriate to note which of the coauthors did the quantification, using initials.

2nd Revision - authors' response

24 November 2016

Authors made requested editorial changes.

Corresponding Author Name: Michael F Olson

Manuscript Number: EMM-2016-06743